# A Variationist Study of Subject Pronoun Expression in Medellín, Colombia

**Rafael Orozco** [1,*] and **Luz Marcela Hurtado** [2]

[1] Department of Foreign Languages and Literatures, Louisiana State University, Baton Rouge, LA 70803, USA

[2] Department of World Languages and Cultures, Central Michigan University, Mount Pleasant, MI 48859, USA; hurta1lm@cmich.edu

\* Correspondence: rafael.orozco@nyu.edu or rorozc1@lsu.edu

**Abstract:** This variationist study of subject pronoun expression (SPE) in Medellín, Colombia uses multivariate regressions to probe the effects of ten predictors on 4623 tokens from the Proyecto para el Estudio Sociolingüístico del Español de España y de América (PRESEEA) corpus. We implement analytical innovations by exploring transitivity and the lexical effect of the verb, which we analyze by testing infinitives and subject pronoun + verb collocations, respectively, as standalone, random-effect factors. Our results reveal the highest pronominal rate (28%) found in a mainland Spanish-speaking community. Additionally, we uncover that pronominal rates increase with age, a finding which appears to have cognitive implications. The internal conditioning contributes to *pronombrista* studies by showing the effects of discourse type and transitivity. Narratives and opinion statements favor overt subjects, but statements indicating routine activities favor null subjects. Whereas unergative verbs promote overt subjects, reflexive verbs favor null subjects. The lexical effect of the verb reveals opposing tendencies between verbs in the same category as well as within different collocations of the same verb, providing more definitive answers than the semantically guided approaches used for the last four decades and showing that verb groupings do not constitute functional categories with regard to SPE. Overall, this study contributes to expand our baseline knowledge of SPE in mainland Latin American communities and opens interesting research avenues.

**Keywords:** subject pronoun expression; Colombia; Colombian Spanish; Latin American Spanish; Andean Spanish; Medellín; sociolinguistics; language variation; lexical effects





## 1. Introduction

The variable alternation between overt and null pronominal subjects (e.g., *tú cantas* alternating with *cantas* to mean 'you sing') constitutes a Spanish morphosyntactic feature inherited from Latin. As it has continued to evolve, Latin appears to have been changing towards becoming less pro-drop, and the typical evolution in the Romance languages seems to move from pro-drop to non-pro-drop. Although Spanish remains a pro-drop or variable subject expression language, Brazilian Portuguese has become a semi-pro-drop language (Erker and Guy 2012, p. 531). Furthermore, Modern French and Haitian Creole are obligatory pronominal subject languages (Leroux and Jarmasz 2005; Ortiz López 2011).

Variationist subject pronoun expression (SPE), i.e., *pronombrista* investigations were pioneered by Barrenechea and Alonso (1973), Morales (1980), and Bentivoglio (1980), with their studies of the Spanish spoken in Buenos Aires, Argentina; San Juan, Puerto Rico; and Caracas, Venezuela, respectively. Those seminal studies inspired the development of a sizeable body of literature that has explored the alternation between null and overt pronominal subjects in Caribbean Spanish (Alfaraz 2015; Bentivoglio 1987; de Prada Pérez 2020; de Prada Pérez and Soler 2020; Orozco 2015a, 2018a; Ortiz López 2009; among others), Mainland Latin American Spanish (Cerrón-Palomino 2014; Lastra and Martín Butragueño 2015; Travis 2005a, 2005b), Peninsular Spanish (Cameron 1993; Enríquez 1984; Posio 2011; de Prada Pérez 2009, 2015), and Spanish in the United States (Cameron 1992, 1995, 1996,

1998; Cameron and Flores-Ferrán 2004; Flores-Ferrán 2002, 2004, 2007a; Hurtado 2001; Limerick 2018; Otheguy and Zentella 2007, 2012; Orozco 2018b; Silva-Corvalán 1982, 1994, 1997; Torres Cacoullos and Travis 2018, and others). Interestingly, *pronombrista* studies have made important contributions to scholarship on Spanish in the United States as well as to variationist sociolinguistics in general.

The vast body of literature exploring SPE throughout the Hispanic World and beyond includes several investigations that provide a foundation for further study of this linguistic variable in Colombian Spanish. Previous *pronombrista* research on Colombian Spanish explored SPE among Colombians in Florida as well as in Bogotá (Hurtado 2001, 2005a, 2005b), in the city of Cali (Travis 2005b, 2007; Travis and Torres Cacoullos 2012), and among Mainlander Colombians in New York City (Otheguy and Zentella 2007, 2012; Otheguy et al. 2007). Subsequent research (Orozco and Guy 2008; Orozco 2015a, 2018a; Hurtado and Ortega-Santos 2019) has studied SPE in Barranquilla and among *Costeño* Colombians in NYC (Orozco 2018a, 2018b). More recent work (De la Rosa 2020) analyzes SPE in Cartagena and San Basilio de Palenque. Nevertheless, SPE remains understudied in Andean Colombian Spanish, despite this variety being spoken by most Colombians. Thus, with the first *pronombrista* study of Medellín Spanish, we seek to fill a knowledge void and open new research paths. Our analysis departs from the traditional exploration of the effects of the verb by means of semantically based categories, as has been done by numerous scholars (Carvalho and Child 2011; Enríquez 1984; Hurtado 2005b; Otheguy and Zentella 2012; Travis 2007; among others). Instead, we explore the verb using transitivity, a syntactically guided verb classification. Our analytical approach is motivated by recent findings revealing statistically significant opposing tendencies between verbs in the same semantically based category. For instance, among verbs of motion, *ir* 'go' favors overt subjects whereas *llevar* 'take, deliver' favors null subjects in Barranquilla, Colombia (Orozco 2018a, p. 116). Additionally, we explore the lexical effect of the verb with the goal of gaining a more detailed understanding of how verbs condition SPE. In the sections that follow, we provide a brief overview of *pronombrismo*, describe our analytical procedure, as well as present and discuss our results.

## 2. Pronombrismo

While Barrenechea and Alonso (1973) included all pronouns in their seminal study, Morales (1980) and Bentivoglio (1980) focused exclusively on the first person singular. As *pronombrista* studies developed, most researchers followed the former pattern although many others explored the first person. Regardless of whether researchers have explored all pronouns or only the first person singular, *pronombrista* research has determined that variable SPE displays notable regional differences in terms of overt pronominal rates and that overall frequency of use differs dialectally. The highest overall overt pronominal expression rates are found in the Caribbean (average 38%), where they range from 33% (Cuban newcomers to New York City, Otheguy and Zentella 2012) to 45% (San Juan, Puerto Rico, Cameron 1993). Concurrently, lower pronominal rates occur in Spain (21%, Enríquez 1984; Cameron 1993) and mainland Latin American varieties such as those of Mexico City (21.8%, Lastra and Martín Butragueño 2015) and Lima, Peru (16.8%, Cerrón-Palomino 2014), with an average of 24%.

Despite well-known overt pronominal rate differences between speech communities in different dialect regions, five decades of *pronombrista* scholarship have uncovered much uniformity regarding both the predictors that probabilistically condition SPE and the tendencies exhibited by their individual factors (Carvalho et al. 2015, p. xiii). The remarkable similarity of effects found lends support to the notion that structured linguistic variation constitutes an intrinsic part of our grammatical knowledge; i.e., usage patterns are deeply embedded in our knowledge of grammar. Overall, SPE is mainly conditioned by grammatical person and number of the subject, coreference; priming; tense, mood, and aspect (TMA) morphology; lexical semantics or verb type; clause type; and verbal reflexivity. Thus, overt pronominal subjects occur more frequently with singular subjects, following a change in

referent, with verb tenses that have ambiguous person morphology such as the imperfect, immediately after an overt subject, and in main clauses. When a reflexive pronoun is used with the same verb, subject pronouns are disfavored. Previous research exploring verb semantics has revealed that cognitive and psychological verbs (e.g., *saber* 'know,' *recordar* 'remember,' *creer* 'believe') promote overt pronominal subjects whereas external activity verbs (e.g., *ir* 'go,' *salir* 'leave,' *trabajar* 'work') promote null subjects (Abreu 2009; Bentivoglio 1980, 1987; Enríquez 1984; Hurtado 2005b; Orozco and Guy 2008; Posio 2011; Travis 2007; among others). High pronominal expression with psychological verbs has been explained as an indicator of the speaker's stance towards the utterance (Travis 2007, p. 117). Moreover, the distinction between the subjectivity and objectivity of the action points out the influence of the subjectification of discourse. This idea is also supported by the fact that singular pronouns (especially the first-person singular and *uno*) have higher pronominal rates and probabilistic weights of expression across Spanish varieties.

In this context, variable subject pronoun expression is related to the speaker's presence in the discourse. Aijón Oliva and Serrano (2010, 2013) and Serrano (2012) propose to analyze the role of agency in the subjectification of discourse, as it is related to the concepts of stance and informativeness: an overt subject is more prominent but less informative than an omitted one (Aijón Oliva and Serrano 2013, p. 311). To analyze the role of agency in SPE, they suggest considering Hopper and Thompson's (1980) components of transitivity and their relationship with the focus of attention, understood as the prominence of the speaker in the events. Studies by Posio (2011) and Hurtado and Ortega-Santos (2019) found correlations between low transitivity verbs and overt realizations of first and second person in Peninsular Spanish and *uno* 'one' in Colombian Spanish (Barranquilla), respectively. That is why, besides including predictors drawn from the large body of work on subject expression, we include verbal transitivity in our analysis to further explore this correlation to all personal pronouns. We also intend to determine whether this predictor can help to better account for pronominal expression, or, on the contrary, the null or overt realizations unveil the contribution of discursive functions (stance, focus of attention, and subjectification) in the semantic-syntactic flexibility of verbs (García-Miguel 2005, p. 172)

### 2.1. Previous Treatments of the Effect of the Verb on SPE

Researchers have explored the effects of the verb on SPE for four decades employing several different perspectives. For example, Silva-Corvalán (1982) separates verbs syntactically according to ambiguous conjugation forms (e.g., *first- and third-person singular verbal endings coincide in the conditional, preterit imperfect,* and *various subjunctive tenses*) and non-ambiguous conjugation forms (e.g., *preterit* and *present indicative*). From a syntactic-semantic perspective, the verb's effect on SPE has been explored by grouping verbal infinitives into such classifications as verb type, lexical content of the verb, and psychological vs. other verbs. These verb classifications are described below.

### 2.1.1. Verb Type

Bentivoglio (1980) was the first to explore the effect of the verb on SPE. Using semantic criteria, she categorized verbs into five types:

a.   cognitive verbs (*pensar* 'think,' *saber* 'know,' *creer* 'believe,' etc.)
b.   perceptive verbs (*oler* 'smell,' *ver* 'see,' *sentir* 'feel,' etc.)
c.   enunciative verbs (*afirmar* 'state;' *decir* 'say, tell;' *comentar* 'comment;' *hablar* 'speak;' etc.)
d.   desiderative and manipulative verbs (*desear* 'wish,' *ordenar* 'command,' *querer* 'want,' *pedir* 'ask,' etc.).
e.   other verbs (i.e., verbs that do not correspond to the above categories).

Silva-Corvalán (1988) incorporates motion verbs to this classification (*andar* 'walk,' *caminar* 'walk,' *corer* 'run,' etc.). This classification has been implemented—in some cases with minor adaptations—in many studies which have shown the conditioning effect of the verb on SPE (e.g., Cerrón-Palomino 2014; Erker and Guy 2012; Hurtado 2001, 2005b; Orozco 2015a, 2018a, 2018b; Travis 2005a, 2005b, 2007; among others). General verb type

tendencies show that cognitive and perceptive verbs favor overt subjects (Silva-Corvalán and Enrique-Arias 2017, p. 175) whereas all other verbs—especially desiderative and other verbs—favor null subjects.

### 2.1.2. Lexical Content of the Verb

Enríquez (1984) originally employed the lexical content of the verb in her study of SPE in Madrid. According to their lexical content, verbs are divided into four groups:

a.  Mental activity verbs, which require psychological effort from the speaker (*acordarse* 'remember,' *analizar* 'analyze,' *aprender* 'learn,' *decidir* 'decide,' *desear* 'wish,' *escoger* 'select,' *imaginar* 'imagine,' *intentar* 'attempt,' etc.).

b.  Estimative verbs, which imply the speaker's opinion or judgment (*admirar* 'admire,' *considerar* 'consider,' *creer* 'believe,' *opinar* 'believe,' *pensar* 'think,' *respetar* 'respect,' *suponer* 'suppose,' etc.).

c.  Stative verbs, which escape all dynamic processes and are not affiliated with activities that the speaker undertakes, either mentally or physically (*crecer* 'grow,' *criarse* 'be raised,' *enamorarse* 'fall in love,' *estar* 'be,' *llamarse* 'be called,' *ser* 'be,' *tener* 'have,' *valer* 'be worth,' *vivir* 'live,' etc.).

d.  External activity verbs, which imply physical, mental, or behavioral activity or which derive from movement, or an ongoing situation (*acabar* 'finish;' *avanzar* 'advance;' *cambiar* 'change;' *comprar* 'buy;' *conseguir* 'obtain, get;' *criar* 'raise;' *dar* 'give;' *decir* 'say, tell;' *entrar* 'enter;' *escribir* 'write;' *hablar* 'speak;' *hacer* 'do, make;' *ir* 'go;' *llegar* 'arrive, come;' *mirar* 'watch;' *salir* 'leave;' *ver* 'see;' etc.).

Since its inception into *pronombrismo* by Enríquez (1984), the lexical content of the verb classification has been used by many scholars, most notably Otheguy and Zentella (2007, 2012) in their monumental study of SPE in New York City Spanish. The detailed lexical content description provided by Otheguy and Zentella (2012) in their SPE coding manual—a *pronombrista* guidebook which they generously shared ahead of its publication— prompted many other researchers (Carvalho and Child 2011; Erker and Guy 2012; Flores-Ferrán 2002, 2004, 2009; Orozco 2015a, 2016, 2018a, 2018b; Orozco and Guy 2008; among others) to explore the effect of verbs on SPE using this classification. Overall, the lexical content of the verb tendencies tell us that (a) stative verbs favor overt subjects; (b) both mental activity verbs and estimative verbs moderately favor overt subjects—this explains why many studies have examined these two types of verbs jointly (e.g., Erker and Guy 2012; Otheguy and Zentella 2012; Orozco and Guy 2008; among others); and (c) external activity verbs favor null subjects.

### 2.1.3. Psychological vs. Other Verbs

In view of the tendencies found when exploring the effect of the verb on SPE by using the semantic categories described above, Torres Cacoullos and Travis (2011) establish a binary classification. According to results using this dualistic arrangement, psychological verbs favor overt subjects, whereas all the other verbs favor null subjects.

### 2.1.4. Lexical Frequency

With the advent of more sophisticated quantitative tools, the use of categories based on lexical or semantic criteria to investigate the effects of the verb on SPE has been considered problematic (Posio 2011, p. 780). Many recent theoretical proposals, such as those following the usage-based and exemplar models (Bybee 2001; Pierrehumbert 2001; Bybee and Torres Cacoullos 2008), postulate that speakers store detailed and extensive information about frequent words and expressions. According to these models, speakers retain abundant information about lexical frequency. Erker and Guy (2012, p. 528) show that, in these models, lexical frequency associates to a conservatory and situational cognitive effect (Bybee et al. 1997; Bybee 2010). In fact, the need to obtain more definitive information about how the verb conditions variable pronominal subject expression has been previously

indicated (Orozco and Guy 2008, p. 77), and a way to obtain this information may be by analyzing verb frequency within the corpus (Erker and Guy 2012; Posio 2011, 2015).

## 3. Materials and Methods

This section provides a description of the speech community studied here. It also describes the corpus and the dataset analyzed. Subsequently, we present the research questions and the hypothesis that guide this analysis. Finally, we discuss the predictors explored, the envelope of variation, and the analytical procedure.

### 3.1. The Speech Community, the Corpus, and the Dataset

Medellín, founded in 1675, has constituted one of Colombia's main industrial centers since the first half of the 20th century. Between 1890 and 1950, the process of textile industrialization and the production of tobacco, beer, ceramics, glass, and coffee promoted an increment of a blue-collar class as well as the city's urbanistic growth (Botero 1996, pp. 8–10). According to the 2018 census, from its population of 2,372,330, 59.36% were born in the city, 37.27% were born elsewhere in Colombia, and 2.2% in another country, which reflects the situation of displacement and migration to the city suffered by Colombians because of violence and lack of economic opportunities (Departamento Administrativo Nacional de Estadística (DANE); Castañeda 2005, p. 82).

The variety of Spanish spoken in Medellín belongs to what Montes Giraldo (1982) classified as Western Andean Colombian Spanish. The city is located in the department of Antioquia, where two dialectal varieties have been identified: the Andean and the Coastal. Because of its geographical location, Medellín developed as an isolated city with differentiated traditions, religiosity, and family values (Fernández Acosta 2020, p. 95). In terms of the Andean variety, this region is characterized by the extensive use of *vos* "you" across ages, social levels, and registers, which has been analyzed not only as an indicator of an egalitarian and open society (Montes Giraldo 1967, p. 25) but also as an expression of local identity and belonging to the region (Fernández Acosta 2020, p. 97; Millán 2014).

The dataset examined here is part of the 119 oral sociolinguistic interviews that constitute the PRESEEA-Medellín corpus (González-Rátiva 2008). This corpus is part of the international Project for the Sociolinguistic Study of Spanish in Spain and the Americas. It was collected considering uniform proportions and representativity in terms of gender, age, education, and socioeconomic levels. Participants belonging to lower socioeconomic levels completed elementary education and lived in densely populated sectors of the city (northeastern area). Participants with secondary-level education resided in medium density areas (northwestern, central-eastern, and central-western), while professionals and university students were located mostly in low-density areas such as the southeast and the southwest (Andrade Rodríguez et al. 2008). From this corpus, we analyzed 39 interviews collected between 2007 and 2010, containing 29 hours of recorded speech. Participants were stratified according to gender, age, and degree of instruction. In this study, we include 20 women and 19 men whose ages ranged from 15 to 85 years old at the time they were interviewed. All of the participants were born in Medellín or in the surrounding region.

The PRESEEA-Medellín corpus (González-Rátiva 2008) is an important part of the online resources for sociolinguistic research provided by Universidad de Antioquia's Department of Communications. It includes the transcription of the interviews following PRESEEA's transcription protocols. These interviews contain important cultural and sociolinguistic information about the people, customs, and life in the city, a valuable resource for improving social conditions in Medellín. The interviews were carried out using topics previously prepared as part of a questionnaire intended to produce semidirected sociolinguistic conversations. Following pragmatic patterns prevalent in the local speech community (Millán 2014, p. 99), the predominant address form used by the interviewers was *usted* (you formal). The questions deal with a variety of topics, ranging from the weather, the neighborhood, Medellín's people, problems in the city, and transportation to more personal questions about family, work, daily routines, customs and holidays,

and narratives of a dream or a scary event. The interviews ended with these narrations. According to the script used by the fieldworkers, the first five minutes were spent on a warm-up to the conversation before starting the semidirected 45-min interview. However, the resulting interviews were of varying lengths and word counts. We extracted the tokens for our analysis starting immediately after the fifth minute of each recording.

*3.2. Research Questions and Hypothesis*

This investigation contributes to a growing baseline of data on contemporary SPE in monolingual Spanish speaking communities. In exploring the constraints that affect the alternation between overt and null pronominal subjects in the Spanish of Medellín, we seek to answer three main research questions. These questions are guided by the findings of numerous previous studies including most of those cited in the preceding sections.

(a) *How are overt and null pronominal subjects distributed in the Spanish of Medellín, and how does this variety compare with other varieties of Spanish in terms of subject pronoun expression?*

(b) *How do social predictors condition SPE in Medellín, and how do their effects in this speech community compare to those in other communities?*

(c) *Is the internal conditioning on subject pronoun expression in Medellín Spanish similar to what is found throughout the Hispanic World? Do all verbs within the same category similarly condition SPE?*

Concurrently, we aim to probe the following main hypothesis: *Despite an internal conditioning concurrent with what is found across the board, different verbs within a single category condition SPE differently*. This hypothesis was informed by studies indicating that we lack conclusive information as to the effects of the verb on SPE (Erker and Guy 2012; Orozco 2015a; Orozco et al. 2014; Orozco and Guy 2008; Posio 2011; inter alia). Additionally, we test a series of hypotheses that directly address each one of the predictors explored here which are discussed below.

*3.3. Predictors Explored*

To answer the above research questions and test our hypotheses, we explored the effects of three social and seven linguistic predictors. We based our choice of predictors on the findings of a multitude of previous *pronombrista* investigations (Cameron 1992, 1993, 1995; Enríquez 1984; Flores-Ferrán 2002, 2004, 2007a; Lastra and Martín Butragueño 2015; Orozco 2015a; Otheguy and Zentella 2007, 2012; Otheguy et al. 2007; Torres Cacoullos and Travis 2011, 2018; Travis 2005a, 2005b, 2007; Travis and Torres Cacoullos 2012; among others). The predictors analyzed here are discussed below.

3.3.1. Social Predictors

The analysis of social predictors constitutes one of the pillars of variationist sociolinguistics. In line with studies that report the conditioning effects of social predictors on SPE (Alfaraz 2015; Lastra and Martín Butragueño 2015; Otheguy and Zentella 2012; de Prada Pérez 2015; Shin and Otheguy 2013; among others), our analysis explores the effects of three social predictors (age, gender, and education) which are considered the main determinants of functions or social roles in language variation (Chambers 2009; Tagliamonte 2012; Silva-Corvalán and Enrique-Arias 2017; among others).

a. Speaker's Age

We probed the effect of speaker's age in this analysis considering that (a) the role of age is one of the cornerstones of variationist research; (b) prior studies have reported the conditioning effect of age on SPE in monolingual Spanish-speaking communities (Orozco and Guy 2008; Lastra and Martín Butragueño 2015; Orozco 2015a; de Prada Pérez 2015; Shin and Erker 2015; among others); and (c) SPE provides a singular opportunity to compare differences in the probabilistic grammars of children and adolescents compared to those of older speakers (Posio 2018, p. 298). Thus, in light of recent findings uncovering that in monolingual speech communities pronominal rates, developmentally, increase with

age (Orozco 2016; Shin 2015; Shin and Erker 2015; among others), we hypothesized that the youngest speakers in our analysis would promote null subjects and that the oldest speakers would promote overt pronominal subjects. Concurrently, given that researchers continue to work on determining the developmental stages at which SPE variability is developed and acquired (Posio 2018, p. 298) and that different SPE studies divide age groups differently (e.g., Alfaraz 2015; Lastra and Martín Butragueño 2015; Orozco 2015a, 2016; among others), we started our analysis of this predictor by establishing the age group configuration that best fits the characteristics of the Medellín speech community, a task that constituted a singular challenge. During our first coding step, we listed each speaker's age on a single column of our coding spreadsheet. After an initial distributional analysis of our data sample, we arrived at the age group distribution that best fits the SPE developmental stages. Thus, we divided our speakers into the following three age groups: (a) 15–29, (b) 30–54, and (c) over 55 years old.

b.　Speaker's Gender

　　Gender differences have constituted an important component of variationist scholarship since its origin (Cheshire 2004, p. 423; Fasold 1990, p. 92). Nevertheless, the pioneering *pronombrista* studies that analyze gender (Bentivoglio 1980, 1987; Cameron 1992; among others) did not find it to significantly condition pronominal expression. Given interesting gender differences found in SPE and the expressions of futurity and possession in Colombian Spanish (Orozco 2007, 2009, 2015b, 2018a) as well as in *pronombrista* studies that report women favoring overt pronominal subjects (Alfaraz 2015; Hurtado 2001, 2005a; among others), we tested the effect of gender in the present study. Additionally, considering the gender effect found by Otheguy and Zentella (2012) and subsequently established and designated as a women effect by Shin (2013) and Shin and Otheguy (2013), we hypothesized that gender would condition SPE in Medellín. Furthermore, guided by findings in Santo Domingo, Dominican Republic (Alfaraz 2015); Barranquilla, Colombia (Orozco 2015a, 2018b); the Colombian community in New York City (Orozco 2018a, 2018b); and Spain (de Prada Pérez 2015), among others, we hypothesized that women would favor overt pronominal subjects.

c.　Education

　　The conditioning effect of education on the expressions of futurity and possession in Colombian Spanish (Orozco 2018a) motivated its inclusion in this analysis. We divided speakers according to the three categories used in the PRESEEA corpus. We hypothesized that those speakers with higher educational attainment would favor null subjects and that those with lower educational attainment would favor overt subjects.

3.3.2. Linguistic Predictors

　　The seven linguistic predictors explored in this analysis operate at three morphosyntactic levels (the whole clause, the subject, and the predicate). The seven internal predictors analyzed here are (1) discourse type; (2) coreferentiality; (3) priming; (4) grammatical person and number of the subject; (5) transitivity (verb class); (6) verbal tense, mood, and aspect (TMA); and (7) lexical effect of the verb.

a.　Discourse Type

　　We analyzed the impact of various discursive modes to measure their effect on pronominal expression as well as the high incidence of first-person subjects: narrative (personal experiences), opinion (argumentative discourse about the city and its people), hypothetical situations (projected actions and contrary to fact actions), description (physical descriptions of their mates, family, friends, the preparation of a meal), and routine.

b.　Coreferentiality

　　Previous studies on Colombian Spanish showed a correlation between overt expression and a total change of reference as well as between null subjects and same previous number and person: Orozco (2018a, p. 106) in Barranquilla's *Costeño* Colombian variety (.67 and .35) and New York (.62 and .39); Hurtado (2005b, pp. 343–44) in the *Costeño* (.60 and .36)

and Andean (.62 and .34) varieties of Colombians living in Miami. This predictor explores the hypothesis that overt pronominal subjects compensate the change in informational situation motivated by a change of refence. Our hypothesis is based on the premise of compensation, which is based on Hochberg's (1986) functional hypothesis. To explore the relationship between the subject under analysis and the immediately preceding subject, we used a different perspective from that which has traditionally been analyzed in *pronombrista* studies (Bentivoglio 1980; Cameron 1995; Lastra and Martín Butragueño 2015; Otheguy and Zentella 2012; Silva-Corvalán 1977, 1994; inter alios). We coded coreferentiality using the following four factors. (1) same grammatical number and same grammatical person, (2) same number but different person, (3) different number but same person, and (4) different number and different person. This approach facilitates the analysis of cases considered complicated by Cameron (1995, p. 17) such as partial coreference or reference overlap between the target and trigger NPs when the target represents a set of which the trigger is a member.

c.  Priming

This predictor explores the possibility that the occurrence of a prior overt or null subject triggers further pronoun expression or omission (Cameron and Flores-Ferrán 2004), as illustrated in these examples:

1.  *Por la timidez que **yo mantenía, yo sudaba** hasta frío para hablarle a ella; es que **yo era** una persona muy tímida.* (H21-5, 489)
    'Because of the shyness that I used to maintain, I had a cold sweat even as I talked to her; it's because I was a very shy person.'
2.  *Ø Creo, Ø no conozco mucho el proyecto, y en lo que Ø conozco, es como el despelote que ha causado en las vías. Pero Ø no sé, Ø no conozco pues tampoco. Ø Sé que es una cosa similar al Transmilenio de Bogotá, pero Ø no estoy tampoco como muy informada* (M13-5, 283).
    'I think I don't know much about the project, and in what I know, it's like the mess that it has caused on the roads. But I don't know, and I'm not familiar with it either. I know that it is something similar to Bogota's Transmilenio, but I'm not that well informed either.'

Priming is a linguistic construct based on the idea that speakers pattern a clause on the preceding discourse in an unintentional way (Travis 2007). Studies on Colombian Spanish (Travis 2005b, 2007; Torres Cacoullos and Travis 2019), and Peninsular and Puerto Rican Spanish (Cameron 1994; Flores-Ferrán 2002) have found significant effects of this variable in SPE and its intersections with coreferentiality, distance from the preceding coreferential subject, and type of discourse. We aim to test if this predictor exerts the same tendencies in the Medellín variety.

d.  Grammatical Person and Number of the Subject

The factors included in this predictor were first-person singular (*yo*), second-person singular (*usted*, *tú*, and *vos*), third-person singular (*él*/*ella* and *uno*), and plural persons (*nosotros* and *ellos*/*ellas*). For the sake of comparison with previous studies, such as in Orozco's (2018a) work on SPE by Colombians living in Barranquilla and in New York, the impersonal *uno* was coded within the third person. Impersonal uses of *uno* and the second person were included in the codification because, as Gómez Torrego (1992) and Fernández (2013) stated, they are considered an extension of the deixis, a metonymic extension through which the subjects represent others. They are also specified in the PRESEEA guidelines. Finally, grouping all plural pronouns under the same factor was a methodological decision because all of them presented similar effects in preliminary results, and *ellos*/*ellas* 'they' had a low token count (3% of the total). We hypothesized that singular grammatical persons, especially *yo* 'I' and *uno* 'one', would favor overt pronominal subjects.

e.  Transitivity (Verb class)

As previous classifications based on lexical content of the verb and verb semantics have proven to be problematic, and because there are still no definitive answers about their effect on subject expression (Posio 2011, p. 780; Orozco 2018a, p. 113), we decided to explore the influence of verb transitivity. We included the verb classes from Hurtado

and Ortega-Santos's (2019, p. 46) study of the null and overt expression of coreferential *uno*, which were based on Hopper and Thompson's (1980) analysis of transitivity. The distinction between transitive (ditransitive, monotransitive, monotransitive with null object, verb with a prepositional complement, and reflexive) and intransitive (unergative and unaccusative) verbs allows us to analyze the degree of competition between the object and the subject for the focus of attention. This classification enables us to observe the influence of the degree of transitivity due to the varying number of participants (subject–object). According to Hopper and Thompson (1980, p. 252), clauses with two or more participants are high in transitivity, and a decrease in the number of participants involves a decrease in transitivity. As seen in example 3, the likelihood of the subject being realized is smaller for the monotransitive verb *llevar* 'to lead' because such a competition is present (2 participants). However, reflexive verbs such *divertirse* 'to enjoy oneself' constitute a case in which the verb might have transitive syntax, yet the number of participants is smaller, being lower in the transitive scale.

3. *Pues ∅ **llevo** una vida muy así tranquila. Sí, pues, ∅ **me divierto** por ahí un rato.* (M21-1, 267)
   'Well, I lead such a quiet life. Yes, so I enjoy myself around a little bit.'

Intransitive verbs have the same number of participants (1), but the agentive subject of the unergatives (e.g., *vivir* 'to live') is more dynamic (Enghels 2012, p. 50) than in the unaccusatives. Still, both are lower in transitivity, so the probability that overt pronouns are used is greater, as in examples 4 and 5.

4. *A ver. El sitio donde **yo vivía** se llama el barrio Restrepo.* (M23-3, 42)
   'Let's see. The place where I lived is called the Restrepo quarter.'
5. *En la casa, lo que más hace uno, pues, **uno llega** del trabajo es a descansar.* (H32-1, 129)
   'At home, what one does more, so, one returns from work only to rest.'

After observing low token counts for ditransitives (5.6%), monotransitives with null object (6.1%), and verbs with a prepositional complement (2.1%), as well as similar tendencies between these factors in preliminary results, we regrouped the factors in the following way: transitives (ditransitives and monotransitives), other (monotransitives with null object and verbs with a prepositional complement), reflexives, unaccusatives, and unergatives. The application of a theoretical framework that combined the concepts of salience and transitivity (Hopper and Thompson 1980) in subject expression in Spanish was first proposed by Aijón Oliva and Serrano (2010) and Serrano (2012), respectively. According to Serrano (2012, p. 111), an overt subject reflects a major cognitive prominence, which is also related to the degree of agency that the expression involves. Posio (2011), for his part, analyzed the effect of transitivity in first- and second-person subject expression and identified a correlation between low transitivity and subject pronoun usage. We hypothesized that this correlation can be extended to all subject pronouns.

f. Verbal Tense, Mood, and Aspect (TMA)

Based on Orozco (2018a, p. 108), we tested the effect of the present, imperfect, and preterit as standalone factors. Additionally, we grouped under "other" the conditional, perfect tenses, subjunctives, futures, and imperatives. Previous studies on subject pronoun variability have shown that imperfective actions favor overt pronouns, possibly because of their ambiguous verbal morphology (Orozco 2015a, p. 26). Thus, the inclusion of this variable is relevant to test the effect of transitivity, as it is related to telic and atelic actions, one of the transitivity components of Hopper and Thompson (1980). As in Hurtado and Ortega-Santos (2019), the prediction is that atelic actions will favor the use of overt pronouns.

g. Lexical Effect of the Verb

In light of previous findings on the verb's effects on SPE (Erker and Guy 2012; Posio 2015, 2018; Orozco 2016, 2018a, 2019, n.d.), we tested the verb as a random-effects predictor with infinitives serving as standalone factors. Although this analytical innovation has been previously called a lexical frequency effect (Orozco 2016, 2018b) due to having a lexical frequency component, what we measure is rather a lexical effect. In fact, these

tendencies do not relate to frequency, as the most frequent verbs do not exert tendencies different from those of the less frequent verbs. Thus, whereas all of the most frequently occurring verbs do not consistently favor overt pronominal subjects, neither do the less frequent verbs consistently favor null subjects. With the purpose of providing a critical look at the traditional metalinguistic categories used to explore the effect of the verb on SPE and besides probing the lexical effects of verbal infinitives, our analysis tests the effect of pronominal subject + verb collocations. These collocations or prefabricated units include all of the tokens in our analysis. Thus, they include both preposed and postposed pronominal subjects, as discussed below in Section 3.4. Collocations also include cases with intervening elements such as *Yo también conozco muchas partes* 'I also know many places.' Further, the intervening elements include negatives, clitics, etc. In probing the lexical effect of the verb, we hypothesized that a more detailed examination of its effects on SPE would show that the traditional categories used for four decades do not fully account for the conditioning effect of verbs on this linguistic variable.

### 3.4. The Envelope of Variation and the Analysis

The envelope of variation used in our analysis adheres to the Principle of Accountability (Labov 1972, p. 72). Additionally, it follows the comprehensive parameters defined by Barrenechea and Alonso (1973), Otheguy and Zentella (2012, p. 48 ff.), and the PRESEEA project (Silva-Corvalán and Enrique-Arias 2017, p. 173 ff.) which are regarded as standard for *pronombrista* studies. We included in our analysis only those clauses with ascertainable animate pronominal subjects that contain a conjugated verb where the alternation between a null and an overt subject is clearly possible. The tokens comprise cases of preposed pronouns (e.g., *Yo iba a montar cicla* 'I would go to ride bikes.') as well as cases of postposed pronouns (*Me mantenía yo en la casa* 'I stayed at home.'). Concomitantly, we excluded meteorological (e.g., *nevar* 'to snow,' and *llover* 'to rain') and existential verbs (e.g., *haber* 'be, exist') as well as fillers (e.g., *digamos* 'let's say'). Thus, all tokens constitute one of at least two possible different ways of saying the same thing.

The data sample used in this study is comprised of 4623 tokens. We coded all tokens in terms of the predictors discussed above on Excel spreadsheets as comma-separated values. Subsequently, we conducted a series of quantitative analyses using Rbrul and Language Variation Suite (Scrivner and Díaz-Campos 2016) as our statistical tools. We started the quantitative exploration of our data sample with a distributional analysis (Tagliamonte 2006, p. 193; 2012, p. 121) and proceeded with crosstabulations intended to detect factor interactions. Among other things, the distributional analysis helped us reconfigure several predictors: case in point, age, given that we had initially considered dividing speakers into age decades (20s, 30s, 40s, and so on). As indicated above (Speaker's Age, Section 3.3.1), we determined that the best configuration fit for our data was by dividing speakers into three age groups: (a) 15–29, (b) 30–54, (c) over 55 years old. Subsequently, we conducted a series of multivariate statistical regression analyses intended to probe the different hypotheses pertaining to each of the predictors discussed above. Preliminary statistical regressions led us to reconfigure several predictors according to token numbers and tendencies throughout subsequent runs.

The following are the three main reconfigurations made to our linguistic predictors. First, we reconfigured discourse type by merging our 31 recipe description tokens with our original 324 general description tokens (given that they both provide descriptions and had neutral tendencies) into our new descriptive statements factor (355 tokens). Second, we reconfigured grammatical person and number of the subject by (a) merging the three second person singular pronouns (*tú, usted,* and *vos,* 'singular you'); (b) merging the three third person singular pronouns (*ella* 'she,' *él* 'he,' and *uno* 'one'); and (c) merging together all plural pronouns (*nosotros* 'masculine we,' *nosotras* 'feminine we,' *ustedes* 'plural you,' *ellas* 'feminine they,' and *ellos* 'masculine they'). Third, we reconfigured verb transitivity by merging (a) monotransitive and ditransitive verbs into transitives and (b) verbs with null

objects and those with prepositional objects. We conducted several rounds of multivariate tests until we arrived at the model presented in the next section.

## 4. Results

In the sections that follow, as we walk the reader through our results, we begin by setting forth the distribution of overt and null pronominal subjects. Our discussion of the social conditioning on pronominal usage precedes that of internal, linguistic constraints. We subsequently draw conclusions and formulate their implications.

### 4.1. Distribution of Variable Pronominal Subjects and Predictor Model

The distribution of overt and null pronominal subjects is presented in Table 1. Medellín's overall 28% overt pronominal rate is significantly lower ($X^2$ = 33.57; $p$ < .001) than Barranquilla's 34% (Orozco 2015a). However, it constitutes the highest overt pronominal rate ever found in a monolingual mainland speech community, as mainland pronominal rates average 24% (Lastra and Martín Butragueño 2015; Michnowicz 2015; Orozco and Guy 2008; Otheguy and Zentella 2007, 2012). One reason for this relatively high pronominal rate may be Medellín's geographical location in a department with a coastal region and in close proximity to Caribbean varieties, which are known for their high pronominal rates.

**Table 1.** Distribution of overt and null subjects.

| Grammatical Person | Pronominal Rate | Overt SPPs | Null Subjects | N | % Data |
| --- | --- | --- | --- | --- | --- |
| 1st singular | 32% | 816 | 1743 | 2559 | 55% |
| 2nd singular | 33% | 67 | 137 | 204 | 4% |
| 3rd singular | 42% | 297 | 417 | 714 | 15% |
| All plural | 10% | 114 | 1032 | 1146 | 25% |
| All pronouns | 28% | 1294 | 3329 | 4623 | 100% |

The distribution of overt and null subjects with its attendant pronominal rate by grammatical person (Table 1 and Figure 1) corroborates the conclusion drawn from the results of previous studies (Abreu 2009, 2012; Bayley and Pease-Alvarez 1997; Bentivoglio 1987, pp. 36, 60; Carvalho and Child 2011; Claes 2011; Erker and Guy 2012; Flores-Ferrán 2002, 2004, 2007b, 2009; Otheguy and Zentella 2007, 2012; Otheguy et al. 2007; Posio 2011; de Prada Pérez 2009; Ortiz López 2011; among others) that singular pronominal subjects occur more frequently as overt subjects than their plural counterparts. It also shows a higher incidence of first-person subjects (55% of the data). Additionally, third-person singular subjects register the highest pronominal rate (42%) whereas that for plural pronouns (10%) is considerably lower than the 28% for the Medellín speech community as a whole.

Initial multivariate results revealed interaction between coreferentiality and priming. Thus, we retained priming for all subsequent multivariate runs. Likewise, to avoid skewed results, we did not test both transitivity and the lexical effect of the verb in the same multivariate run. The results of several rounds of multivariate tests uncovered a complex model that includes linguistic and social forces with six predictors (one social and five linguistic) reaching statistical significance (See Table 2). The order of selection shows grammatical person and number of the subject as the strongest predictor with a $p$-value of $4.05^{-69}$. In general, internal constraints have a greater conditioning effect on SPE based on their order of selection, which was established according to $p$-values. The greater internal conditioning found in Medellín is consonant with what obtains throughout the Spanish-speaking world (Carvalho et al. 2015).

**Figure 1.** Mosaic plot of overt and null pronominal subjects by grammatical person.

**Table 2.** The Medellín subject pronoun expression (SPE) model.

| Predictor | *p*-Value | Range |
|---|---|---|
| Grammatical person and number of the subject | $4.05^{-69}$ | 42 |
| Discourse type | $2.91^{-20}$ | 40 |
| Priming | $6.27^{-19}$ | 17 |
| Tense Mood & Aspect | $8.91^{-11}$ | 23 |
| Speaker's age | $1.62^{-8}$ | 12 |
| Transitivity | $2.15^{-5}$ | 18 |

Moreover, the constraint hierarchy found in Medellín with (a) grammatical person and number of the subject and (b) priming (prior subject's realization) being among the strongest internal predictors is largely consonant with findings around the Hispanic World including Barranquilla, Colombia (Orozco 2015a); Los Angeles (Silva-Corvalán 1982, 1997); Madrid, (Enríquez 1984); Mexico City (Lastra and Martín Butragueño 2015); Puerto Rico (Cameron 1993, 1995); New York City (Otheguy and Zentella 2007, 2012), Rivera, Uruguay (Carvalho and Bessett 2015); and Yucatan, Mexico (Michnowicz 2015); inter alia. This finding corroborates that, despite varying pronominal rates at the surface level, the grammar underlying SPE across varieties remains essentially the same (Cameron 1993; Michnowicz 2015; Travis 2007; Torres Cacoullos and Travis 2011). Our presentation of the results of the effects of the predictors conditioning SPE in Medellín follows. In our presentation, we preserved the same order in which the predictors examined here were discussed. Thus, we first address the social predictors and subsequently the internal conditioning.

*4.2. Social Conditioning*

As stated above (Section 3.3.1), our study explores three social predictors: speaker's age, education, and gender. Our findings reveal that while age significantly constrains SPE in Medellín Spanish, speaker's gender and education do not. The results for gender (Table 3) show that both men and women register the same probability levels (.50) and pronominal rates (28%). Compared with other *pronombrista* research, the lack of significance of gender in Medellín is congruent with what obtains in Caracas (Bentivoglio 1980, 1987), Mexico City (Lastra and Martín Butragueño 2015), Yucatán (Michnowicz 2015), and the Uruguay-Brazil border region (Carvalho and Bessett 2015), among other speech communities, where women and men display similar SPE tendencies. At the same time, the lack of significance of gender in Medellín differs from what occurs in other Colombian speech communities as well as from what happens elsewhere in the Hispanic World, given

that a sizable body of work has found overt pronominal subjects to be favored by women (e.g., Bayley and Pease-Alvarez 1996; Solomon 1999; Carvalho and Child 2011; Otheguy and Zentella 2012; Shin and Otheguy 2013; Alfaraz 2015; Orozco 2015a, 2018b). These results disprove our hypotheses that gender would condition SPE in Medellín with women favoring overt pronominal subjects, as they suggest the existence of different gender effects for SPE in different speech communities.

**Table 3.** Social conditioning on SPE in Medellín, Colombia.

| Factor | Prob. | % Overt | N | *p*-Value |
|---|---|---|---|---|
| *Speaker's Gender* | | | | |
| Women | [.50] | 28% | 2357 | |
| Men | [.50] | 28% | 2266 | |
| *Range* | *0* | | | N.S. |
| *Speaker's Age* | | | | |
| Over 55 | **.57** | 33% | 1687 | |
| 30 to 54 | .48 | 27% | 1839 | |
| 15 to 29 | .45 | 24% | 1097 | |
| *Range* | *12* | | | $1.62^{-8}$ |

The results for age, also presented in Table 3, uncover that speakers over the age of 55 favor overt pronominal subjects with a probability of .57; middle-aged speakers (30 to 54 years old) have a neutral effect (.48); and the youngest segment of the population, i.e., speakers younger than 30, favor null subjects (.45). That is, our results reveal that pronominal rates proportionally increase with age. These findings support our hypothesis that our oldest speakers promote overt pronominal subjects whereas our youngest speakers favor null subjects. Concomitantly, our results are consonant with what occurs in other monolingual speech communities including Barranquilla, Colombia (Orozco and Guy 2008; Orozco 2015a); Mexico City (Lastra and Martín Butragueño 2015); Oaxaca, Mexico (Shin and Erker 2015); and Spain (de Prada Pérez 2015); among others.

*4.3. Internal Conditioning*

The internal conditioning on SPE in Medellín is presented in Table 4. It reveals the effects of two predictors pertaining to the subject (grammatical person and number of the subject and priming), one predictor pertaining to the whole clause (discourse type), and two verb-related predictors (verb transitivity and TMA). Our presentation of the results pertaining to internal predictors will follow that same order. Thus, we first deal with grammatical person and number of the subject and will close with those pertaining to the lexical effect of the verb.

4.3.1. Subject-Related Conditioning

a. Grammatical Person and Number of the Subject

As mentioned before, Medellín adheres to the general tendency across varieties of Spanish with singular pronouns favoring overt subjects and plural pronouns disfavoring them. However, Table 4 reflects an idiosyncratic feature of Medellín speech, as the third person singular is the factor that most strongly promotes overt pronominal subjects with a statistical weight of .64. An explanation for this result can be found in the fact that this classification includes *él/ella* 'he/she' and *uno* 'one,' whose overt pronominal rates are 26.6% and 60%, respectively. Hurtado and Ortega-Santos's (2019, p. 51) study on the use of *uno* in Barranquilla, Colombia already reported the dominance of overt coreferential *uno* over its null coreferential counterpart. Nevertheless, these results differ from Orozco's previous studies on SPE by Colombians living in Barranquilla and New York, respectively (Orozco 2018a, p. 107), in which the third person (*él/ella* and *uno*) favored overt pronouns (Barranquilla .61, NY .58) but not as strongly as the first person (Barranquilla .68, NY

.64). Either way, as *uno* is used with the purpose not only of diminishing the agent's responsibility but also of placing the speaker's perspective at the forefront (Company Company and Loyo 2009, p. 1196), what these results suggest is a connection between pronominal expression and the *yo/uno*-speaker. The percentage of overt use of *uno* when the speaker refers to personal experiences (68%) allows us to suggest that it fulfills the function of the expression of the speaker's stance, more so than constituting a reduction of prominence, as illustrated in example 6:

6.  *Entonces **yo asumí** toda la responsabilidad de mi casa. Cuando **Ø empecé** a trabajar, pues ya uno con su platica, pues entonces uno ya con un ambiente distinto, uno como padre en la casa, entonces **uno se sentía** pues que **Ø era** pues como muy grande, ¿cierto? Entonces, en ese tiempo, siempre **Ø me tomaba** mis guaros.* (H33-3, 318-323)
    'So, I assumed all the responsibility of my household. When I started to work, then one with one's money, so then one is in a different environment, one as a parent in the household; thus, one felt, well that . . . was something very big, right? So, at that time, I would always drink my shots.'

Results for the first and second person showed an identical favoring effect in subject expression (.59). Interestingly, the use of *usted* prevails as the preferred form of address in this variety of Colombian Spanish. From the 204 cases of second-person singular subjects (Table 4), 160 were instances of *usted* and 44 were of *vos* and *tú*.

**Table 4.** Internal conditioning on SPE in Medellín, Colombia.

| Factor | Prob. | % Overt | N | *p*-Value |
|---|---|---|---|---|
| *Grammatical Person and Number of the Subject* | | | | |
| Third Singular | **.64** | 42% | 714 | |
| Second Singular | .59 | 33% | 204 | |
| First Singular | .59 | 32% | 2559 | |
| Plural | .22 | 10% | 1146 | |
| *Range* | *42* | | | $4.05^{-69}$ |
| *Discourse Type* | | | | |
| Opinion | **.62** | 33% | 1326 | |
| Narrative | .60 | 29% | 2033 | |
| Hypothetical Situation | .58 | 28% | 560 | |
| Description | .51 | 25% | 355 | |
| Routine | .22 | 7% | 349 | |
| *Range* | *40* | | | $2.91^{-20}$ |
| *Tense Mood and Aspect* | | | | |
| Imperfect | **.61** | 37% | 422 | |
| Present Indicative | .53 | 29% | 2832 | |
| Preterite Indicative | .49 | 26% | 563 | |
| Other | .38 | 21% | 806 | |
| *Range* | *23* | | | $8.91^{-11}$ |
| *Verb Transitivity* | | | | |
| Unergative | **.61** | 35% | 305 | |
| Transitive | .52 | 31% | 2194 | |
| Unaccusative | .51 | 27% | 1059 | |
| Other | .44 | 26% | 377 | |
| Reflexive | .43 | 19% | 688 | |
| *Range* | *18* | | | $2.15^{-5}$ |
| *Priming* | | | | |
| Previous Overt Subject | **.56** | 40% | 970 | |
| Outside Priming Environment | .55 | 35% | 1024 | |
| Previous Null Subject | .39 | 21% | 2629 | |
| *Range* | *17* | | | $6.27^{-19}$ |

b.  Priming

The effect of this predictor is demonstrated as we found that a previous overt subject favors subsequent overt subjects with a probabilistic weight of .56. Conversely, a priming effect is also established by the low probabilistic weight (.39) of SPP realizations after a previous null subject. Our Medellín findings are congruent with *Costeño* Spanish (Orozco 2018a, p. 104), as the statistical weights in Barranquilla and New York showed the same tendencies: prior overt subject pronouns promoted the occurrence of overt SPPs (.60 and .61); prior null occurrences favored null subjects (.43 and .42). Also, in the Colombian variety of Cali (Torres Cacoullos and Travis 2019, p. 671), null subjects are favored by previous unexpressed subjects (.63) but disfavored by a previous overt pronoun (.37). Therefore, regional dialect is not a relevant factor in terms of priming in Colombian Spanish, neither is it in other Spanish varieties such as Madrid, Puerto Ricans living in San Juan and in New York (Cameron and Flores-Ferrán 2004), and even in other languages (Meyerhoff 2009).

### 4.3.2. Clause-Related Conditioning: Discourse Type

As Table 4 shows, type of discourse exerted the second strongest influence over pronominal expression, with opinion, narrative, and hypothetical situations as factors that favored explicit subject pronouns (.62, .60, and .58) and routine as the factor that disfavored them (.22). These results clearly indicate the link between pronominal expression and speaker's stance and experiences (as seen in example 6 above). Lastra and Martín Butragueño (2015) also found a favoring effect of argumentation (.66) in Mexico City, which they connected to positioning points of view and highlighting opinions.

Another result that suggests a relationship between pronominal usage and discursive genre is the neutral effect of description (.51), which is not focused on the speaker (example 7). Most interview questions elicited descriptions of a third person singular or plural, whose pronominal expression rates were 26.6% and 13.4%, respectively.

7.  *No, **ella es** bien, ∅ **es** muy alegre, **ella es** chévere. **Ella tiene** raticos que, pues ∅ **es aburrida**, y hay ratos que ∅ **es** alegre. Siempre ∅ **ha sido** pues, buena mujer*. (H11-4, 214)
    'No, she is well (good), she is very happy, she is awesome. She's got her moments, well she's boring, and she's got times when she's happy. She's always been well, a good woman."

Likewise, the influence of pronominal verbs on null subject production indicates the importance of the verbal function, as reflexive verbs were extensively used by the speakers to talk about their daily routines, which is consistent with previous studies (Abreu 2009, 2012; Bayley and Pease-Alvarez 1997; Carvalho and Child 2011; Otheguy and Zentella 2012):

8.  *Pues la rutina mía como diaria, pues ∅ **me levanto**, o sea, todos los días ∅ **tengo** clase, si no es de seis es de ocho, ∅ **me levanto,** pues ∅ **me baño**, normal, ∅ **me voy** para la universidad.* (M03-5, 349-354)
    'Well, my daily routine, so I get up, well, every day I have class; if it's not at six, it's at eight. I get up, then I bathe; normally, I go to the university.'

As we will observe in Section 4.3.3, the disfavoring effect of the discourse on routine is related to the disfavoring effect of reflexive verbs.

### 4.3.3. Verb-Related Conditioning

a.  Verb Tense, Mood, and Aspect (TMA)

Results show that the imperfect tense favors overt pronominal expression (.61); the present with a probability value of .53 moderately favors overt subjects, and the preterit has a neutral effect (.49). At the same time, all other tenses, acting as a single factor, favor null pronouns (.38). The same direction of influence is also found by Orozco (2018a, p. 109) in the Barranquilla *Costeño* Colombian variety (imperfect .61, present .52, preterit .47, and others .40) and in New York (imperfect .58, present .56, preterit .47, and others .38). Our results are also consonant with findings in other monolingual communities (Bentivoglio

1987; Enríquez 1984; Travis 2007; Lastra and Martín Butragueño 2015; Carvalho and Bessett 2015; among others).

These tendencies provide partial evidence for the idea of the influence of ambiguous forms on subject expression, as the imperfect is the tense with the highest probabilistic weight. However, other ambiguous forms such as the conditional and subjunctives (included in "other") do not favor overt pronouns. The present and the preterit (unambiguous forms) have little or neutral effect, and the cases of the imperfect represent only 9% of the data set.

b.  Verb Transitivity

In Table 4, factor weights show that intransitive unergatives (.61) promote overt pronominal subjects in contrast to "other" (monotransitives with null objects and verbs with prepositional complements .44) and reflexive transitive verbs (.43). So far, the high probabilistic weight for unergatives corroborates that low transitivity increases subject expression. However, the fact that unaccusatives and transitives show a more neutral effect (.51 and .53) could imply that it is not categorical transitivity per se what affects the distribution of pronominal subjects but rather a gradient transitivity on the focus of attention (Posio 2011; Hurtado and Ortega-Santos 2019), null/overt object distinction, and other flexibilities in similar verbs. Similar tendencies are shared in the Peninsular and Barranquilla varieties. In Peninsular Spanish, overt first- and second-person singular were disfavored with agentive verbs with object arguments, which were likely the focus of attention (e.g., *dar* 'give'), but favored with one-argument verbs such as stative verbs (e.g., *ser* 'be'), because the subject is usually the focus of attention (Posio 2011, p. 796). In Barranquilla, unergatives and transitives with a null object also correlated with the use of overt coreferent *uno* 'one,' in contrast to non-reflexive monotransitives and ditransitive verbs (Hurtado and Ortega-Santos 2019, p. 51).

c.  Lexical Effect of the Verb

We probed the lexical effect of the verb with the goal of providing an alternative to the metalinguistic categories traditionally used to explore the effect of the verb on SPE. The results presented in Table 5, in line with our analytical objectives, include only those verbs that occur with a pronominal subject either overt or null. Although our lexical effects analysis includes the infinitive form of all the 374 different verbs that occur in our dataset, the results presented in Table 5 include only the 30 most frequent verbs in SPE contexts; that is, those which constitute at least 0.5% of the data by occurring 24 or more times. In general terms, the most frequent verbs can be divided into three groups according to their tendencies:

(a)  verbs that favor overt pronominal subjects (*creer* 'think, believe;' *pensar* 'think;' *decir* 'say, tell;' *vivir* 'live;' *trabajar* 'work;' etc.)

(b)  verbs with a neutral effect (*tener* 'have,' *ir* 'go,' *estar* 'be,' *poder* 'can,' *quedar* 'stay,' and *irse* 'leave'), and

(c)  verbs that favor null subjects (*poner* 'put;' *imaginarse* 'imagine;' *volver* 'return;' *mirar* 'look;' *hablar* 'speak, talk;' *venir* 'come;' *llevar* 'take, carry').

The findings in Table 5 show that *tener* 'have,' the most frequent verb in our data with 425 occurrences, has a neutral effect with a probability value of .510. *Ser* 'be,' the second most frequent verb with 341 occurrences, favors overt pronominal subjects (.625), and *hablar* 'speak, talk,' our third most frequent verb with 221 cases, disfavors overt subjects (.432). On the other hand, we find among less frequent verbs that *acordarse* 'remember' (N 24) promotes overt pronominal subjects (.618); *quedar* 'stay' (N 43) has a neutral effect (.504); and *poner* 'put' (N 38) strongly promotes null subjects (.328). In other words, the most frequent verbs do not consistently exert tendencies different from those of the less frequent verbs. Consequently, we possess mounting evidence indicating that a lexical frequency effect, in itself, does not condition pronominal subject expression.

Moreover, our results provide a detailed account of the verb's effects on SPE. They also reveal information previously obscured by the (semantically based) classifications

traditionally employed in *pronombrista* studies to explore the effects of the verb (Bentivoglio 1980; Enríquez 1984; Orozco 2018b; among others). From that perspective, *creer* 'believe,' and *pensar* 'think' with probability values of .882 and .747, respectively, appear to cause the favorable effect of cognitive or mental activity verbs on overt pronominal subjects. At the same time, *volver* 'return' (.397) and *poner* 'put' (328) appear to account for the favorable effect of motion and external activity verbs on null subjects.

**Table 5.** Lexical effect of the verb.

| Factor | Prob. | N | % Overt | % Data |
|---|---|---|---|---|
| *Creer* 'believe' | .882 | 155 | 72.3% | 3.4% |
| *Pensar* 'think' | .747 | 84 | 50.0% | 1.8% |
| *Decir* 'say, tell' | .737 | 195 | 46.7% | 4.2% |
| *Vivir* 'live' | .728 | 103 | 46.6% | 2.2% |
| *Trabajar* 'work' | .723 | 47 | 48.9% | 1.0% |
| *Comprar* 'buy' | .646 | 27 | 40.7% | 0.6% |
| *Ser* 'be' | .625 | 341 | 33.4% | 7.4% |
| *Llegar* 'arrive' | .618 | 61 | 34.4% | 1.3% |
| *Acordarse* 'remember' | .618 | 24 | 37.5% | 0.5% |
| *Conocer* 'know' | .602 | 84 | 32.1% | 1.8% |
| *Querer* 'want' | .587 | 78 | 30.8% | 1.7% |
| *Saber* 'know' | .572 | 216 | 28.7% | 4.7% |
| *Ver* 'see' | .566 | 170 | 28.2% | 3.7% |
| *Levantarse* 'get up' | .550 | 28 | 28.6% | 0.6% |
| *Salir* 'exit, leave' | .533 | 66 | 25.8% | 1.4% |
| *Tener* 'have' | .510 | 425 | 23.5% | 9.2% |
| *Ir* 'go' | .508 | 132 | 23.5% | 2.9% |
| *Estar* 'be' | .506 | 219 | 23.3% | 4.7% |
| *Poder* 'can' | .504 | 108 | 23.1% | 2.3% |
| *Quedar* 'stay' | .504 | 43 | 23.3% | 0.9% |
| *Irse* 'leave' | .479 | 71 | 21.1% | 1.5% |
| *Sentir* 'feel' | .466 | 60 | 20.0% | 1.3% |
| *Pasar* 'pass' | .465 | 26 | 19.2% | 0.6% |
| *Llevar* 'take, carry' | .458 | 32 | 18.8% | 0.7% |
| *Hablar* 'speak, talk' | .432 | 221 | 18.1% | 4.8% |
| *Venir* 'come' | .407 | 53 | 15.1% | 1.1% |
| *Mirar* 'look' | .402 | 36 | 13.9% | 0.8% |
| *Volver* 'return' | .397 | 31 | 12.9% | 0.7% |
| *Imaginarse* 'imagine' | .375 | 48 | 12.5% | 1.0% |
| *Poner* 'put' | .328 | 38 | 7.9% | 0.8% |

As discussed in the previous section, our analysis shows that transitivity, a syntactically based classification, conditions pronominal expression. Thus, we used our lexical effects analysis to probe whether transitivity provides a better alternative to explore the effects of the verb than semantically based classifications. Our lexical effect findings uncover that although transitivity provides a novel way to explore the effect of the verb on SPE, and it significantly conditions pronominal expression, it also seems to obscure opposing tendencies between verbs within a given category. That is, not all verbs in a given transitivity category exert similar tendencies.

Our probe of the transitivity categories reveals that among monotransitive verbs *creer* 'believe;' *pensar* 'think;' and *decir* 'say, tell' strongly favor overt subjects. Contrarywise, *mirar* 'look' and *poner* 'put' favor null subjects, but *tener* 'have' has a neutral effect. Among unergative verbs, *vivir* 'live' favors overt subjects, *poder* 'can' has a neutral effect, and *hablar* 'speak, talk' disfavors overt subjects. The findings for unaccusative verbs show that *llegar* 'arrive' favors overt pronominal subjects, but *volver* 'return' favors null subjects while *salir* 'exit, leave' has a neutral effect. The tendencies for reflexive verbs reveal that *acordarse* 'remember' and *levantarse* 'get up' favor overt subjects although *irse* 'leave' and *imaginarse* 'imagine' disfavor them.

Table 6 reports the most significant opposing tendencies within verbs of the same transitivity category. These discrepancies were tested for statistical significance by means of $X^2$ tests. In fact, the greatest discrepancy found within a given transitivity category is that between *creer* 'believe' (.882, pronominal rate 72%) and *poner* 'put' (.328, pronominal rate 8%).

**Table 6.** Comparisons between verbs within a given lexical category.

| Verb | Prob. | N | % Overt | Category | $X^2$ | $p$ |
|---|---|---|---|---|---|---|
| *Creer* 'believe' | .882 | 112/155 | 72.3% | Monotransitive | 49.86 | <.001 |
| *Poner* 'put' | .328 | 3/38 | 7.9% | | | |
| *Vivir* 'live' | .728 | 48/103 | 46.6% | Unergative | 27.43 | <.001 |
| *Hablar* 'speak, talk' | .432 | 40/221 | 18.1% | | | |
| *Llegar* 'arrive' | .618 | 21/61 | 34.4% | Unaccusative | 3.79 | .05 |
| *Volver* 'return' | .397 | 4/31 | 12.9% | | | |
| *Acordarse* 'remember' | .618 | 9/24 | 37.5% | Reflexive | 4.64 | .03 |
| *Imaginarse* 'imagine' | .375 | 6/48 | 12.5% | | | |

Taking our analysis one step further, we compared the lexical effect in Medellín to what obtains in other speech communities. Interestingly, one fact that emerged was the strong effect of *creer* 'believe' favoring overt pronominal subjects. This concurs with what obtains in other varieties of Colombian Spanish including Cali (Travis and Torres Cacoullos 2012), Barranquilla, and the Colombian community in New York City (Orozco 2018a) as well as in Xalapa Mexico (Orozco 2016). It also concurs with what obtains with the verbs *ser* 'be' and *ir* 'go' in those other three speech communities. Based on what we have seen so far, we could suppose that, as with the internal conditioning on SPE across the board (Carvalho et al. 2015), the lexical effect of the verb is also fairly uniform across varieties of Spanish. However, as our analysis turns to verbs with other pronominal tendencies, we need to adjust our perception, as the following findings emerge.

- *Saber* 'know' has a neutral effect in Medellín and the New York Colombian community and favors overt subjects in Barranquilla but favors null subjects in Xalapa, Mexico.
- *Tener* 'have' has a neutral effect in Medellín and promotes overt subjects in Barranquilla but favors null subjects in Xalapa and New York, respectively.
- *Hacer* 'make, do' despite favoring null subjects in Medellín and Xalapa, has the opposite effect in the New York Colombian community and has a neutral effect in Barranquilla.

We can conclude that unlike the uniformity of effects that has been established for internal predictors across the board over five decades of *pronombrista* research (Carvalho et al. 2015), a clear pattern of verb effects is not discernible. Our analysis of the effects of the verb concurs with that of Orozco (2018a, p. 113 ff.) in producing evidence that a lexical effects analysis uncovers important details including oppositional tendencies between verbs in the same semantic or syntactic category. One important implication of our lexical effects analysis is that it proves a limitation in the usefulness of lexical categories in linguistic research. With the advent of more sophisticated quantitative tools, we have shown that our lexical effect analysis renders semantically or syntactically based lexical categories unable to fully account for lexical effects on language variation and change. Consequently, we are able to better explain the effect of the verb on SPE. Among other things, prior findings regarding the effect of the verb on SPE would need to be revised given that the assumption that all verbs in a given category either promote or disfavor overt subjects has been proven inaccurate. Additional information is found in the Appendix A.

Although our lexical effect analysis using verbal infinitives shows flaws in the traditional verb categories, our analysis of the lexical effects of verbal infinitives continues to group verb forms into one lemma. Thus, we provide an even more detailed analysis by exploring the effects of pronominal subject + verb collocations. The effects of the most

frequent pronominal subject + verb collocations are presented in Table 7. The favorable effect of (*yo*) *creo* 'I think, I believe' stands out. This effect is congruent with findings in Cali, Colombia (Travis and Torres Cacoullos 2012, p. 739), Xalapa, Mexico, and among Central American speakers in Louisiana (Orozco n.d.). Thus, this finding is consonant with *yo creo* 'I think, I believe'—one of the most the frequently occurring SPE collocations—consistently promoting overt subjects (Travis and Torres Cacoullos 2012, p. 739; Torres Cacoullos and Travis 2018) across the board. Aside from the favorable effect of *yo creo* 'I think, I believe' on overt subjects, we can also see that the remaining most-frequently occurring pronominal subject + verb constructions, as with the lexical effect of infinitives, do not seem to exert a frequency effect. That is, the more frequent collocations do not exert tendencies consistently different or opposite to those of the less frequent ones. Therefore, we now possess further evidence that lexical frequency alone cannot account for the effect of the verb on SPE. Concurrently, as Table 7 shows, our findings regarding the effects of pronominal subject + verb collocations also reveal opposing tendencies between inflectional forms of a single verb. Table 7 shows opposing tendencies between finite forms of *tener* 'have' and *ser* 'be.'

**Table 7.** Effects of pronominal subject + verb collocations.

| Collocation | Prob. | % Overt | N | % Data |
|:---:|:---:|:---:|:---:|:---:|
| *Creo* | .877 | 73.0% | 108/148 | 3.2% |
| *Sabe* | .876 | 81.0% | 17/21 | 0.5% |
| *Soy* | .807 | 60.0% | 39/65 | 1.4% |
| *Vivo* | .785 | 63.0% | 17/27 | 0.6% |
| *Tenía* | .770 | 51.6% | 16/31 | 0.67% |
| *Pienso* | .757 | 54.5% | 30/55 | 1.19% |
| *Digo* | .754 | 52.3% | 46/88 | 1.90% |
| *Estaba* | .747 | 51.9% | 14/27 | 0.58% |
| *Ve* | .739 | 53.6% | 15/28 | 0.61% |
| *Dije* | .731 | 52.2% | 12/23 | 0.50% |
| *Puede* | .721 | 45.0% | 9/20 | 0.43% |
| *Es* | .700 | 37.4% | 34/91 | 1.97% |
| *Tiene* | .690 | 41.2% | 14/34 | 0.74% |
| *Está* | .689 | 40.0% | 8/20 | 0.43% |
| *Era* | .669 | 36.7% | 11/30 | 0.65% |
| *Recuerdo* | .593 | 36.8% | 7/19 | 0.41% |
| *Hago* | .591 | 25.6% | 10/39 | 0.84% |
| *Conozco* | .583 | 30.8% | 12/39 | 0.84% |
| *Acuerdo* | .567 | 33.3% | 6/18 | 0.39% |
| *He tenido* | .554 | 26.3% | 5/19 | 0.41% |
| *Conocí* | .542 | 32.0% | 8/25 | 0.54% |
| *Levanto* | .538 | 26.1% | 6/23 | 0.50% |
| *Voy* | .538 | 25.0% | 7/28 | 0.61% |
| *Tengo* | .530 | 25.0% | 34/136 | 2.94% |
| *Veo* | .521 | 27.8% | 15/54 | 1.17% |
| *Somos* | .500 | 23.5% | 8/34 | 0.74% |
| *Salgo* | .495 | 20.0% | 5/25 | 0.54% |
| *Siento* | .492 | 22.2% | 8/36 | 0.78% |
| *Me voy* | .468 | 13.6% | 3/22 | 0.48% |
| *Sé* | .468 | 22.4% | 34/152 | 3.29% |
| *Puedo* | .431 | 15.8% | 3/19 | 0.41% |
| *Estoy* | .421 | 18.0% | 11/61 | 1.32% |
| *Imagino* | .384 | 13.2% | 5/38 | 0.82% |
| *Eramos* | .382 | 14.3% | 3/21 | 0.45% |
| *Tenemos* | .350 | 12.7% | 8/63 | 1.36% |
| *Son* | .323 | 10.3% | 3/29 | 0.63% |
| *Vea* | .300 | 7.4% | 2/27 | 0.58% |
| *Estamos* | .267 | 5.9% | 2/34 | 0.74% |
| *Hacemos* | .250 | 3.6% | 1/28 | 0.61% |
| *Vamos* | .227 | 0.0% | 0/23 | 0.50% |

Our pronominal subject + verb collocations analytical innovation leads to the conclusion that these constructions constitute two kinds of prefabs in Spanish: *(yo) creo* 'I think, I believe' and all others. *(Yo) creo* 'I think, I believe' stands out as a unit that appears to have been cognitively reanalyzed and become grammaticalized as a discourse formula (Torres Cacoullos and Walker 2009). Thus, it consistently favors overt pronominal subjects cross-dialectally (Travis and Torres Cacoullos 2012) and exemplifies an instance of *autonomy* (Bybee 2003, 2006; Hopper and Traugott 2003, p. 127). At the same time, all other pronominal subject + verb collocations remain grammatically productive. In other words, on the one hand, *yo creo* 'I think, I believe' consistently favors overt pronominal subjects across the board. On the other hand, all other subject + verb collocations appear to have different tendencies in different speech communities. Thus, although the effect of the verb on SPE was considered a resolved issue with cognitive-psych verbs promoting overt subjects (Carvalho et al. 2015, p. xv; Linford and Shin 2013; Orozco 2015a; inter alias), we now have evidence that the semantically-based classifications used to explore the effect of the verb on SPE for the last four decades (Bentivoglio 1980; Enríquez 1984) fail to uncover important differences not only between verbs in a given category but between inflectional forms of the same verb.

Moreover, by exploring the verb in terms of pronominal subject + verb collocations (Bybee 2010; Bybee and Eddington 2006; Bybee and Torres Cacoullos 2008; Croft and Cruse 2004; Travis and Torres Cacoullos 2012; among others), our analysis is more detailed than previous analyses and uncovers important facts, including numerous instances of opposing tendencies regarding finite forms of a single verb. These findings suggest that exploring the effects of the verb on SPE by using collocations informs our collective knowledge beyond what we already know. Thus, it appears that by using analyses which probe the effects of pronominal subject + verb collocations (Bybee and Eddington 2006; Bybee and Torres Cacoullos 2008; Goldberg 2006; Torres Cacoullos and Walker 2009; Travis and Torres Cacoullos 2012; among others), we can obtain more conclusive answers as to how lexical effects condition language variation and change.

## 5. Discussion

Our *pronombrista* study of Medellín Spanish has addressed three research questions and a main hypothesis. The answer to our first research question (*How are overt and null pronominal subjects distributed in the Spanish of Medellín, and how does this variety compare with other* varieties *of Spanish in terms of subject pronoun expression?*) reveals an overt pronominal rate of 28%. Although this pronominal rate is representative of a mainland, monolingual Spanish-speaking community (Carvalho and Bessett 2015; Lastra and Martín Butragueño 2015; Orozco 2016), it constitutes the highest such rate found in a highland speech community. One reason for Medellín's relatively high pronominal rate seems to be its proximity to the Costeño dialect region. Additionally, migrations and displacement to the city of individuals coming from other parts of Colombia, constituting 37% of Medellín's population, create a dialect contact situation and contribute to account for its pronominal rate. The distribution of overt and null pronominal subjects also reveals that first-person pronominal subjects constitute 55% of the data. Something to be determined by future research is whether the higher incidence of first-person subjects is due to the data-gathering methods employed or it depends on differences in speech genre. In regard to data-gathering methods, sociolinguistic interviews have traditionally asked consultants to describe their life experiences with a setting that, apparently, sets up speakers to use the first person singular more frequently than all other grammatical persons.

The answer to our second research question (*How do social predictors condition SPE in Medellín, and how do their effects in this speech community compare to those in other communities*?) uncovers the significant effect of age, the only social predictor that conditions pronominal expression in our data. That is, neither education nor speaker's gender conditions SPE in Medellín. The age effect found reveals that pronominal rates increase with age, with our youngest speakers favoring null subjects and the oldest speakers promoting overt

pronominal subjects. This is consonant with findings in Barranquilla (Orozco 2015a), Santo Domingo (Alfaraz 2015), and Mexico City (Lastra and Martín Butragueño 2015), among other speech communities. The youngest speakers' disfavoring effect on overt subjects appears to have cognitive and language acquisition implications given that Spanish appears to be changing toward higher pronominal rates, but the youngest speakers are defying expectations (Chambers 2009) by not being the promoters of this change. Instead, our results are explained by the finding that, in monolingual speech communities, pronominal rates increase developmentally with age (Orozco 2016; Shin 2015; Shin and Erker 2015; among others). Thus, native Spanish speakers appear to reach adult pronominal usage in their 20s rather than as young children or teenagers. The lack of significance of gender, on the one hand, concurs with findings in other Mainland speech communities such as Mexico City (Lastra and Martín Butragueño 2015). On the other hand, it is incongruent with what happens in a number of other communities, both monolingual and bilingual, where women promote overt pronominal subjects including Santo Domingo, Dominican Republic (Alfaraz 2015); Barranquilla, Colombia (Orozco 2015a, 2018b); New York City (Otheguy and Zentella 2012; Shin and Otheguy 2013; Orozco 2018a, 2018b); and Spain (de Prada Pérez 2015). Concomitantly, gender conditions other linguistic variables such as the expression of nominal possession in other Colombian communities (Orozco 2018a) as well as in Medellín (Freeman 2019). Thus, our results contribute to provide mounting evidence suggesting that the effect of gender varies with respect to different speech communities and linguistic variables.

The answer to our third research question (*Is the internal conditioning on subject pronoun expression in Medellín Spanish similar to what is found throughout the Hispanic World? Do all verbs within the same category similarly condition SPE?*) reveals a linguistic conditioning similar to what occurs throughout the Spanish-speaking world, as attested in the vast body of *pronombrista* research (Carvalho et al. 2015). In fact, the effects of priming and TMA provide further evidence of one subjacent grammar for all varieties of Spanish regardless of significant pronominal rate differences (Cameron and Flores-Ferrán 2004; Travis 2005b, 2007; Torres Cacoullos and Travis 2019). Still, we only provided partial evidence to the idea of the influence of TMA and morphological ambiguity in SPE, as other ambiguous forms such as conditionals and subjunctives did not favor overt pronouns. In terms of Silva-Corvalán's (1997) remarks, distinctions of discursive functions such as terminative vs. nonterminative actions, realis/irealis forms or backgrounded/foregrounded events, could represent better accounts of this predictor. These functions are consonant with Hopper and Thompson's (1980) components of transitivity such as aspect, punctuality, and mode.

Furthermore, some of our findings also indicate that by analyzing discursive functions we can enhance the explanatory power of internal predictors. We uncovered that coreferential *uno* 'one' and *yo* 'I' exert major influences on the occurrence of overt pronominal subjects, which suggests a connection between pronominal expression and the speaker. Second person singular (mainly *usted*) also showed a favoring effect in subject expression, a possible strategy of the speaker to involve the listener and to establish a relationship with him/her (intersubjectivity in terms of Traugott 2010, pp. 30–33). In regard to the role of discursive types, Company Company and Loyo (2009, p. 1207) propose that *uno* confers certain subjectivity in argumentation, narration, and dialog. Our results for discourse type also unveiled the role of subjectivity, as a discourse genre that focuses on the speaker's experiences and stance (narrative, opinion, and hypothetical situations) favored overt subject expression. This influence also suggests that subjective discourse is low in transitivity and, therefore, favors overt pronominal expression. Concurrently, low transitivity tenses (imperfect of indicative) and low transitivity verbs (unergatives) correlate with the use of overt pronouns, providing further evidence of the pertinence of considering Hopper and Thompson's analysis of transitivity in SPE. However, to be able to explain the fact that pronominal expression was promoted with unergatives and had a neutral effect with unaccusatives, we should resort to the lexical and semantic characteristics of these verbs: the activities and processes of unergative verbs depend on the volitional control of the agent in

contrast to states and non-agentive unaccusative verbs. Thus, pronominal expression also depends on the role of a more or less dynamic and agentive subject (Enghels 2012, p. 51).

Our analysis of verb effects goes beyond previous research (Otheguy and Zentella 2012; Orozco 2015a; Torres Cacoullos and Travis 2018; among others) by exploring the effects of transitivity. Moreover, as has been found in Colombian Costeño Spanish (Orozco 2018a) and Mexican Spanish (Orozco 2016), all verbs within the same category do not exert the same conditioning tendencies. This finding validates our main hypothesis (*Despite an internal conditioning concurrent with what is found across the board, different verbs within a single category condition SPE differently.*). Thus, despite being semantically similar, verbs can differ not only in terms of the syntactic schemes that each verb admits but also regarding their specific conceptual differences. The mixed effect in subject expression within verbs considered under the same semantic or lexical group can be explained if we also consider the degree of transitivity. For example, pronominal expression is promoted by more dynamic agentive subjects as in the case of intransitive-unergative verbs of movement, but the less dynamic intransitive-unaccusative verbs of direction of movement have a neutral effect. Likewise, analyzing the degree of control of the subject over the action allows us to explain the opposite effect in subject expression that we found among the perception verbs *mirar* 'watch' and *ver* 'see'. As Taylor (1995, p. 208) suggests, the act of watching is under the control of the subject (it is more volitional and intentional), and it is more transitive than seeing. Therefore, *mirar* disfavors subject expression because the object is more affected and individuated.

This lexical effect analysis helps respond to recent research that shows the lexical effect of the verb (a) not to depend on lexical frequency; that is, the most frequent verbs do not behave differently from the less frequent ones; and (b) to exert different conditioning effects in different speech communities (Posio 2011, 2015; Orozco 2018a, 2018b; Orozco et al. 2014). That is, recent findings provide mounting evidence that, despite four decades of research, we are yet to know the real effects of the verb on SPE. It appears that the lexical effect of the verb measured by means of specific pronominal subject + verb collocations or prefabs (Bybee 2010; Bybee and Eddington 2006; Bybee and Torres Cacoullos 2008; Croft and Cruse 2004; Goldberg 2006) can help us provide a more detailed account of how verbs condition SPE.

Thus, these results set the verb apart from all other internal language variation and change predictors. They suggest, inter alia, that the differences in how verbs condition language variation in our corpora may be triggered by lexical idiosyncrasy. That is, the lexical effects on a given linguistic variable differ among speech communities due to the intrinsic idiosyncratic characteristics of every community's lexicon. Concomitantly, these findings, by helping to account for the effects of pronominal subject + verb collocations on SPE, have important implications in terms of the relationship between grammaticalization theory and language variation (Bybee 2010; Croft and Cruse 2004; Goldberg 2006). One such implication pertains to the role of collocations such as *yo creo* 'I think, I believe' acting as single grammaticalized units rather than analyzable pronoun + verb combinations, as noted by Travis and Torres Cacoullos (2012, p. 739). Their grammaticalization would, in turn, affect their variation patterns. This study widens our collective analytical scope and enhances the explanatory power of our findings. Our analysis further demonstrates that the current state of affairs regarding lexical effects on SPE and, perhaps, other linguistic variables merits further investigation, as it opens unprecedented inquiry avenues.

## 6. Conclusions

The results of this study contribute to expand the growing baseline of data on SPE in Colombian Spanish. We find that Medellín's overall overt pronominal rate (28%) constitutes the highest such rate ever found among mainland Spanish varieties. Although this pronominal rate is consonant with dialectal classifications that place the *Paisa* variety within the Andean dialect region, it also appears to reflect Medellín's geographical proximity to the Caribbean coast as well as the influx of *Costeño* speakers that has taken place in recent

years. Our variationist analysis also finds an overall conditioning that is, broadly speaking, congruent with the Spanish-speaking world at large, providing further evidence of the stability of the Spanish grammar. Moreover, despite reports that social constraints do not significantly condition SPE in monolingual Spanish varieties (Bayley and Pease-Alvarez 1997; Bentivoglio 1987; Cameron 1992, 1993; Flores-Ferrán 2002; Martínez-Sanz 2011), the present study concurs with Ávila-Jiménez (1995), Alfaraz (2015), Lastra and Martín Butragueño (2015), and Orozco (2015a, 2018a), among others, in finding that social constraints indeed condition pronominal expression. In fact, the robust effect of age uncovers that our youngest speakers (15 to 29 years old), with an overt pronominal rate of 24%, promote null subjects. On the other hand, speakers over 55 years of age promote overt subjects with a pronominal rate of 33%. These tendencies, although perhaps surprising at first sight, are congruent with findings in other speech communities. Additionally, the Medellín age effect corroborates findings indicating that the pronominal rates of children and adolescents increase gradually as they acquire adult SPE usage patterns (Shin 2015, p. 11; Shin and Erker 2015).

The internal conditioning on SPE, with grammatical person and number of the subject as the strongest predictor in Medellín, is largely similar to that throughout the Hispanic World. Nevertheless, despite our findings being consonant with what happens across the board, the tendencies for grammatical person and number of the subject uncover, as presented in Table 4, the highest third person singular pronominal rate in Colombia (42%). This pronominal rate, which is significantly higher than the first-person pronominal rate (32%), appears to be prompted by the proliferation of the impersonal third-person pronoun *uno* 'one' in Medellín. Moreover, our treatment of transitivity constitutes an important contribution to *pronombrista* scholarship, as this is just the third study of the influence of transitivity on pronominal expression (Posio 2011; Hurtado and Ortega-Santos 2019). Here we have expanded the analysis of transitivity on pronominal expression to all grammatical persons, and we have demonstrated through our lexical effect analyses that there are different degrees of conditioning within a single verb classification, being transitive or intransitive. Concurrently, a more detailed analysis of the effects of the verb supports the main hypothesis tested here by revealing that all verbs within a given category do not condition SPE similarly. Specifically, there are discrepancies between verbs in the monotransitive, unergative, unaccusative, and reflexive categories. These oppositional tendencies are best illustrated by the reflexive verbs, with *acordarse* 'remember' favoring overt pronominal subjects but *imaginarse* 'imagine' having the opposite effect. Interestingly, this is also extensive to semantic-based verb classification, as these verbs also fall within the mental activity and cognitive categories. One important implication of our investigation is that it proves a limitation in the usefulness of lexical categories in linguistic analysis. By capitalizing on the availability of more sophisticated quantitative tools and analytical approaches, we have shown that our lexical effects analysis renders semantically based lexical categories unable to fully account for the lexical effects on language variation and change.

In general, this study contributes to enrich our knowledge of SPE. Our results show differences between the effects of both internal and external predictors in Medellín and other communities. Further study shall provide more definitive information regarding the nature of these differences. This study also helps to open new research paths, as it highlights shortcomings in how we have been exploring the effects of the verb on SPE and perhaps other linguistic variables such as the individuation of the object. Moreover, our research contributes to show that we stand to benefit from integrating the analysis of semantic and syntactic predictors to increase the explanatory power of the forces that condition SPE. Among other things, such analytical approach would provide us with a more comprehensive understanding of such phenomena as the effects of the verb, competition for the focus of attention, volition, and intention.

**Author Contributions:** Conceptualization: R.O. and L.M.H.; methodology, R.O. and L.M.H.; software, R.O.; validation, R.O. and L.M.H.; formal analysis, R.O. and L.M.H.; investigation, R.O. and L.M.H.; resources, R.O. and L.M.H.; data curation, R.O. and L.M.H.; writing—original draft preparation, R.O. and L.M.H.; writing—review and editing, R.O. and L.M.H. All authors have read and agreed to the published version of the manuscript.

**Funding:** This research was partially funded by a Louisiana State University 2018 ASPIRE Summer Internship Award granted to Rafael Orozco and Noelle Primeaux.

**Institutional Review Board Statement:** The study was conducted according to the guidelines of the Declaration of Helsinki and approved by the Institutional Review Board of Louisiana State University; approval # E4042; expiration date 14 September 2023.

**Informed Consent Statement:** Informed consent was obtained from all subjects involved in the study.

**Data Availability Statement:** Data were obtained from Proyecto para el Estudio Sociolingüístico del Español de España y de América (PRESEEA) Medellín (http://comunicaciones.udea.edu.co/corpuslinguistico/) The data presented in this study may be made available on request from the corresponding author. The data are not publicly available to accord with the informed consent guidelines provided to the participants.

**Acknowledgments:** We would like to thank Marianne Dieck and María Claudia González-Rátiva for providing us generous access to the PRESSEA-Medellín corpus. We are also thankful to the speakers who provided the data. This research benefited from valuable contributions by the following LSU students who helped with token extraction: Alex Thomas, Alex Yandell, Baily Daberkow, Cecelia Morise, Cecilia Vazquez, Joelle Semplonius, and Sabrina Valenti. We are especially grateful to Noelle Primeaux for her valuable help with data extraction and coding. We are thankful to the audience at IClavE 2019, the anonymous reviewers, and the editors of this volume, Esther Brown and Javier Rivas, for their insightful feedback. We are also grateful to the *Languages* Editorial Office staff for their assistance. All infelicities are our own.

**Conflicts of Interest:** The authors declare no conflict of interest.

## Appendix A

**Table A1.** Lexical effect of the 50 most frequent verbs.

| Factor | Prob. | % Overt | N | % Data |
|---|---|---|---|---|
| *Creer* 'believe' | .882 | 72.3% | 155 | 3.4% |
| *Pensar* 'think' | .747 | 50.0% | 84 | 1.8% |
| *Decir* 'say, tell' | .737 | 46.7% | 195 | 4.2% |
| *Vivir* 'live' | .728 | 46.6% | 103 | 2.2% |
| *Trabajar* 'work' | .723 | 48.9% | 47 | 1.0% |
| *Comprar* 'buy' | .646 | 40.7% | 27 | 0.6% |
| *Recordar* 'remember' | .627 | 40.0% | 20 | 0.4% |
| *Ser* 'be' | .625 | 33.4% | 341 | 7.4% |
| *Llegar* 'arrive' | .618 | 34.4% | 61 | 1.3% |
| *Acordarse* 'remember' | .618 | 37.5% | 24 | 0.5% |
| *Meter* 'stick' | .617 | 38.1% | 21 | 0.5% |
| *Tratar* 'try' | .615 | 38.9% | 18 | 0.4% |
| *Conocer* 'know' | .602 | 32.1% | 84 | 1.8% |
| *Querer* 'want' | .587 | 30.8% | 78 | 1.7% |
| *Saber* 'know' | .572 | 28.7% | 216 | 4.7% |
| *Ver* 'see' | .566 | 28.2% | 170 | 3.7% |
| *Levantarse* 'get up' | .55 | 28.6% | 28 | 0.6% |
| *Salir* 'exit, leave' | .533 | 25.8% | 66 | 1.4% |
| *Tener* 'have' | .510 | 23.5% | 425 | 9.2% |
| *Ir* 'go' | .508 | 23.5% | 132 | 2.9% |
| *Jugar* 'play' | .507 | 23.8% | 21 | 0.5% |
| *Encontrar(se)* 'find, meet' | .507 | 23.8% | 21 | 0.5% |
| *Estar* 'be' | .506 | 23.3% | 219 | 4.7% |
| *Coger* 'take' | .504 | 23.5% | 17 | 0.4% |
| *Poder* 'can' | .504 | 23.1% | 108 | 2.3% |
| *Quedar* 'stay' | .504 | 23.3% | 43 | 0.9% |
| *Ayudarse* 'help' | .484 | 21.1% | 19 | 0.4% |
| *Irse* 'leave' | .479 | 21.1% | 71 | 1.5% |
| *Sentir* 'feel' | .466 | 20.0% | 60 | 1.3% |
| *Pasar* 'pass' | .465 | 19.2% | 26 | 0.6% |
| *Llevar* 'take, carry' | .458 | 18.8% | 32 | 0.7% |
| *Empezar* 'start' | .458 | 17.6% | 17 | 0.4% |
| *Casarse* 'marry' | .449 | 16.7% | 18 | 0.4% |
| *Necesitar* 'need' | .448 | 16.7% | 18 | 0.4% |
| *Reunir(se)* 'meet' | .440 | 15.8% | 19 | 0.4% |
| *Hablar* 'speak, talk' | .432 | 18.1% | 221 | 4.8% |
| *Entender* 'understand' | .431 | 15.0% | 20 | 0.4% |
| *Saludar* 'greet' | .428 | 13.3% | 15 | 0.3% |
| *Bajar* 'get down' | .410 | 11.8% | 17 | 0.4% |
| *Venir* 'come' | .407 | 15.1% | 53 | 1.1% |
| *Mirar* 'look' | .402 | 13.9% | 36 | 0.8% |
| *Volver* 'return' | .397 | 12.9% | 31 | 0.7% |
| *Imaginarse* 'imagine' | .375 | 12.5% | 48 | 1.0% |
| *Poner* 'put' | .328 | 7.9% | 38 | 0.8% |

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
