# Peer review of "A Variationist Study of Subject Pronoun Expression in Medellín, Colombia"

_languages, doi:10.3390/languages6010005_

Round 1
Reviewer 1 Report
The following is a very general overview of certain issues and suggestions for improving the paper. The author should see the PDF document for numerous detailed comments.
This is a very interesting study of variable SPE in Medellin, Colombia. The objectives are clearly explained and the results are well-presented. The findings also reveal fascinating patterns for verb effects and transitivity as they relate to SPE. That said, there are issues with insufficient background/previous studies to provide a relevant context for the current study. The paper goes right from the Intro to the Methods section, and a separate Previous Studies section should be added before talking about the methods. This new section should thoroughly address each of the linguistic and social factors treated in the current study; that is, what have others found concerning these factors?
Additionally, the Methodology section needs to be expanded to include a presentation of the variable context (envelope of variation) and also how each of the factors were operationalized/coded, with examples from the data. Additional information is needed regarding the speakers’ demographics as well.
Results: The factor of “prior subject’s person/number” is mentioned; However, the author does not discuss this factor at all prior to this. Was it included? It does not appear on any of the Tables. Mention of this factor needs to be removed if not included, or clarified if it was in fact included. Likewise, “Coreferentialty” is mentioned as one of the predictors explored (Section 2.3), but is not discussed in the Results. Please clarify this. There are additional issues with the presentation of results, as well as the need to include additional citations. The author will see specific comments and suggestions related to this in the PDF file.

Author Response
We are extremely appreciative for the very insightful comments provided to our paper. We feel that the comments were quite beneficial. In response to the comments provided by the editors and the reviewers, we have made major changes to our manuscript the Microsoft Word Track Changes feature.
During the review process, our paper has been expanded significantly. The first version that you evaluated was 19 pages long. The new version is 27 pages long. In response to the comments received, we added a section (p. 2) titled pronombrismo, where we discuss prior literature. We also expanded Section 3 (Materials and Methods, p. 4 ff.) by adding descriptions of the different predictors explored in our analysis. Additionally, we have justified the inclusion of the linguistic and extralinguistic factor groups that are part of our analysis and we have explained them. We added an analysis of the effect of pronoun + verb collocations on subject pronoun expression. Furthermore, we have responded to all comments.
We will happily provide you with all the additional information you wish to receive.

Reviewer 2 Report
In general, I find this paper interesting and relevant for the state-of-the-art in the research of pronoun use in different varieties of Spanish. The author presents a quantitative analysis of sociolinguistic and discourse variables in guiding overt use of personal pronouns. The research itself makes an interesting contribution to the scientific understanding of the variables that guide over pronominal use in Spanish.
The paper is well-written and does not lack any important part. However, I would like to comment some aspects which I believe could help the author to improve his/her work.
Concerning the content itself of the paper, I would personally very much appreciate if the author could include some very short explanations of why coastal varieties of Spanish are more pronominally-overt. This could clarify, for example, why higher rates of overt pronoun use are found in Medellín, which is argued to be more Carribean like.
As for the data presentation and analysis, I definitely miss information about deviations (SD) in the distribution of overt pronoun use across speakers. Table 1 shows important predominance of overt pronoun use with singular persons, but is it really uniform for all speakers?
There are some content inconsistencies in the text. For example, the abstract speaks about 8 predictors, but the paper itself mentions 6 internal + 3 social (that is, 9) predictors.
On p. 6 the author says: "Compared with other pronombrista research, the lack of significance of sex in Medellín differs from what occurs in other Colombian speech communities as well as from what happens elsewhere in the Hispanic World. It also differs when compared to what happens with other linguistic variables in other Colombian speech communities." What exactly happens with the influence of "sex" on the use of overt pronoun? I think it is really important to mention who use it more in the Hispanic World -men or women- since it implies how the definition of Medellín may be done correctly.
I partially disagree with the coherence of considering "uno" for 3 Singular as equivalente to "he/she". "Uno" is not at all equivalent to "he/she", which are referential, while "uno" may be collective, neutral, impersonal, etc. This disagreement arises also from my surprise that the paper doesn't mention at any moment the pragmatic values of overt pronoun use. Spanish language is particularly pragmatic when using overt pronouns (it may be both referential and mitigating, and can in fact fulfill very many pragmatic functions).
I miss information about "priming" in guiding overt pronoun use. I don't find it convincing. Maybe, the author should include some rationale about why (and to which degree) an overt pronoun may condition further overt pronominal use.
Finally, the author mentions at the beginning of the paper that age variation in the usage of overt pronouns reflect cognitive implications. However, I have not found any further development of this idea in the paper itself.
The first paragraph on the p. 3 is a full repetition of the beginning of the last paragraph on the p. 2. The same is for the last 4 lines of the p. 4, which are repeated. Please, check and delete repetitions.
Please, also check the punctuation of the paper: I have come through unnecessary commas and semicolons in the text (to put an example: "We subsequently, draw conclusions" on p. 4).
Author Response

(The authors gave the same response as above.)

Reviewer 3 Report
Languages Review - A Variationist Study of Subject Pronoun Expression in Medellín, Colombia
This paper offers a noteworthy contribution in its consideration of subject pronoun expression in an understudied region using independent variables that are either absent or operationalized differently in comparable work. The authors clearly have vast expertise on the topic at hand and their use of helpful, contextualized examples is exemplary. Nevertheless, there are a number of ways in which the analysis, organization, and writing can be strengthened, which will enable the authors make a greater contribution and connect to a wider body of research and potentially interested readers. The authors generally show an excellent control of formal, academic English, although I’ve flagged a number of issues in the attached PDF and some of the main comments below.
A lot has been written about SPE, as you note in your introduction, but you don't say very much prior to your results. I would expand your treatment of what is relevant to know from past literature about SPE because as of now, this is rather minimal. Also, you don’t describe your independent variables until the results’ section, when they’re analyzed simultaneously. Most readers are accustomed to a description of variables in the Method section, and then the results for the variables in the subsequent section. It was a bit odd to read results for variables that had not yet been described. Somewhat relatedly, a lot of the interpretation of your results occurs in the results’ section. I would say this is up to the issue editors’ preference as to whether they’d like this reserved for the discussion section or are comfortable with the levels of discussion/interpretation in the results section. Also, I would consider describing your results using the past tense, as is generally the norm, although I’d imagine the editors will indicate whether they share that preference.
Your consideration of verbs as lexemes (i.e., infinitives) rather than how they might differ across word-forms (e.g., person/number/tense) might abscond additional patterns in your data (Tables 5 and A1); see more about this below and the references to the importance of such consideration in Linford and Shin (2013) and the native-speaker data of Linford et al. (2016). Relatedly, some of your discussion of individual verbs misses or oversimplifies competing patterns. For example, different effects for different stative verbs and a different effect for ir than that of other studies (see more detail below). You would also benefit from tying your discussion of lexical effects to similar findings that have been shown for other morphosyntactic variants, such as the copula and mood contrasts. See suggestions below and sources therein.
I like your appeal to Thompson and Hopper’s seminal work on transitivity, although I think the paper would benefit from introducing this notion earlier. Finally, I think you can say more about the inconsistent role of lexical frequency for your verbs, which is an interesting finding absent from the conclusion section.
See below for more of the main comments by line number, along with the attached PDF for numerous, smaller-scale typographical and wording suggestions, along with many positive comments. I commend the authors on their interesting study and hope that these comments are useful. I think that the authors will be able to implement these changes adeptly.
Main comments:
Line 79 “sex” - I would call this gender, regardless of what it's been called previously for these data, since it likely reflects how the participants presented themselves / responded when inquired, rather than necessarily their biological organs. I would change this throughout the paper.
Lines 95-100: You have the exact information of this paragraph in the prior paragraph. Check and streamline.
Line 148 - Statistical test? Or do you mean substantially?
Table 1 - I would add a total column so that it's clear that 32% = 816/X, etc.
Lines 164-172 - This interpretation of rate differences would make more sense in a discussion section than in the initial results.
Figure 1 - I would change your terminology on the X and Y axes to match the terms you use in the body paragraphs. Likewise for other figures.
Table 2 - I would add a range column, since there's some tendency to prefer not associating p-values with strengths of effect. I agree that p-values are appropriate to consider in this sense when talking about regressions, but I think just to be safe with a diverse set of readers, you'll benefit from adding the ranges here.
Lines 183-185: Isn't discourse type stronger than priming? That's an internal factor, too.
Lines 200-202: “It also differs with…” How so?
Table 3 - Put non-significant probabilities in brackets. Check throughout. Tables 3-5 would also benefit from more informative titles.
Lines 225-227: I agree that uno is appropriate to consider within the envelope of variation, but I'd imagine some readers would need some convincing, since it's not a traditional personal pronoun. You might benefit from a sentence or two about this, in text or the footnote, since there isn't much discussion of it here.
Table 4- It seems that there are a number of interactions for verb transitivity, with unaccusative and other being flipped for factor weights and percentages, and ditto for reflexive and monotransitive with null object. I would try cross-tabulating this factor group with other FGs to see if you can figure out the interaction. You also could consider combining some of the similar factors here, especially those with lower token counts.
Line 236 (priming) - It seems odd to not describe the variable at all in the introduction or method, and then explain it for the first time in the results.
Line 254 (discourse type) - Again, seems odd to first introduce the variable and its categories in the results. This appears to apply to the rest of your variables as well, so I'd provide those initial descriptions earlier in the text, and then the results can simply be results.
Lines 261-262 - You vary between present and past tense in this sentence. Traditionally, results are described using the past tense. Check throughout.
Lines 292-295 - This seems like something perhaps for the future directions / limitations discussion rather than the results?
Lines 304 – 310 (transitivity) - Yes, good point. You might also appeal to Hopper and Thompson's seminal work on transitivity here. Although I see it in your reference list, and looks like it will come up in a few paragraphs. But perhaps citing them earlier would be useful, since you're offering their more nuanced-notion of transitivity, as opposed to the binary distinction.
Lines 325-237 - I don't think you mean this, but your wording might -- reword this sentence so that you don't have "reduction for competition" modified by "which increases subject expression," since it's more competition that will increase expression, not reduction of competition.
Section 3.4 - Since verbs occur at different rates with subject pronouns across different persons and numbers, it would be interesting to see these results across different persons/numbers. I agree that as a random effect, it is logical to use the lexeme, as you do. It would also be interesting to see the extent to which these results differ for a particular set of verbs across person and tense, as these can vary widely. For more on this, see Linford and Shin (2013) and the native-speaker data of Linford et al. (2016):
Linford, B., & Shin, N.L. (2013). Lexical frequency effects on L2 Spanish subject pronoun expression. In J. Cabrelli Amaro et al. (eds.), Selected proceedings of the 16th Hispanic Linguistics Symposium, (pp. 175-189). Somerville, MA: Cascadilla Proceedings Project.
Linford, B., Long, A., Solon, M., & Geeslin, K. (2016). Measuring lexical frequency: Comparison groups and subject expression in L2 Spanish. In L. Ortega, A. E. Tyler, H. I. Park, & M. Uno (Eds.), The usage-based study of language learning and multilingualism (pp. 137-154). Washington, DC: Georgetown University Press.
Lines 360-364 – Re: verb type -- Although, for example, the state ser favors overt subjects whereas tener and estar don't particularly favor nor disfavor them.
Table 5 - This range would be written as 55, not 554. In my experience, ranges are not usually provided for a random effect, since the idea is usually that such an effect is something held constant to not influence other variables rather than serving as a constraining effect with an effect size (i.e., range). Perhaps if you've found someone who talks about providing ranges for a random effect, you could cite them here, as this is something I'm not familiar with in research in this vein.
Lines 401-405 - If you want to tie this finding in beyond SPE, similar findings regarding lexical effects (i.e., differential patterning within the same semantic/syntactic categories) are reported for native speakers with respect to the copula contrast and mood variation:
Kanwit, M., & Geeslin, K. L. (2018). Exploring lexical effects in second language interpretation: The case of mood in Spanish adverbial clauses. Studies in Second Language Acquisition, 40, 579–603.
Kanwit, M., & Geeslin, K. (2020). Sociolinguistic competence and interpreting variable structures in a second language: A study of the copula contrast in native and second-language Spanish. Studies in Second Language Acquisition, 42(4), 775-799.
Lines 406-407 - Exactly, and also true for other variable structures as indicated above.
Line 424-426 (re: one underlying grammar across varieties) - Good. Perhaps you can add some citations supporting this?
Lines 431-433 (re: less dynamic verbs showing greater subject expression): Good, but you should acknowledge, on the other hand, that some of the states you analyzed didn't particularly favor overt pronouns, and they would be even less dynamic than the predicates you mention here.
Lines 451-452: Ir's factor weight is .508 and rate of expression is 23.5%, which is below the 28% baseline in your study. I'm not sure that it's fair to say that it favors overt subjects. According to the factor weight, it doesn't particularly favor nor disfavor overt subjects. (And irse is even lower for FW and percentage.)
Line 489 - I believe unergative is the more common term in English?
Lines 480-497 - You might note lexical frequency here, since it can be a motivator of differences within classes, although in your study you show that the verbs did not pattern according to frequency either. Thus, this is another contribution of your study that would be relevant to note here.
Lines 502-506 – You have issues with missing wording in 4-5 places, as indicated in the PDF.

Author Response

(The authors gave the same response as above.)

Round 2
Reviewer 1 Report
The paper has been improved significantly. I very much enjoyed reading the revised version. The author should just make the following minor corrections:
-Top of p. 7: Author writes “six” linguistic Variables, but then lists seven.
Line 432—the phrase “prior subject’s grammatical person and number” should be removed. The author rightly clarified this factor by adding “Priming”, but priming and “prior subject’s person/number” are two completely different factors. Perhaps replace the phrase “prior subject’s grammatical person and number” with “prior subject’s realization as expressed or unexpressed” or something similar.
Author Response
Reviewer 1 Comments:
The paper has been improved significantly. I very much enjoyed reading the revised version. The author should just make the following minor corrections:
Authors’ response: We are very appreciative for the reviewer’s feedback. In response to the Round 2 comments, we have made additional changes to our manuscript.
Reviewer’s comment: -Top of p. 7: Author writes “six” linguistic Variables, but then lists seven.
Authors’ response: We have changed “six” to “seven.”
Reviewer’s comment: Line 432—the phrase “prior subject’s grammatical person and number” should be removed. The author rightly clarified this factor by adding “Priming”, but priming and “prior subject’s person/number” are two completely different factors. Perhaps replace the phrase “prior subject’s grammatical person and number” with “prior subject’s realization as expressed or unexpressed” or something similar.
Authors’ response: We replaced the phrase “prior subject’s grammatical person and number” with “prior subject’s realization.”
We revised the discussion section. Also, we added some references that were missing as well as others that we cited while revising the manuscript. Additionally, we revised the abstract to align it with the revisions implemented throughout the whole manuscript.
We are very appreciative for your valuable contribution to our paper.